# AKT-dependent signaling of extracellular cues through telomeres impact on tumorigenesis

**Raúl Sánchez-Vázquez**[ID], **Paula Martínez**[ID], **Maria A. Blasco**[ID]*

Telomeres and Telomerase Group, Molecular Oncology Program, Spanish National Research Cancer Centre (CNIO), Melchor Fernández Almagro 3, Madrid, Spain

* mblasco@cnio.es

**Data Availability Statement:** All relevant data are within the manuscript and its Supporting Information files.

**Funding:** Research in the Blasco lab is funded by Spanish State Research Agency (AEI), Ministry of

## Abstract

The telomere-bound shelterin complex is essential for chromosome-end protection and genomic stability. Little is known on the regulation of shelterin components by extracellular signals including developmental and environmental cues. Here, we show that human TRF1 is subjected to AKT-dependent regulation. To study the importance of this modification *in vivo*, we generate *knock-in* human cell lines carrying non-phosphorylatable mutants of the AKT-dependent TRF1 phosphorylation sites by CRISPR-Cas9. We find that TRF1 mutant cells show decreased TRF1 binding to telomeres and increased global and telomeric DNA damage. Human cells carrying non-phosphorylatable mutant *TRF1* alleles show accelerated telomere shortening, demonstrating that AKT-dependent TRF1 phosphorylation regulates telomere maintenance *in vivo*. TRF1 mutant cells show an impaired response to proliferative extracellular signals as well as a decreased tumorigenesis potential. These findings indicate that telomere protection and telomere length can be regulated by extracellular signals upstream of PI3K/AKT activation, such as growth factors, nutrients or immune regulators, and this has an impact on tumorigenesis potential.

## Author summary

We show how extracellular milieu information is transmitted to the nucleus through modifications in the telomeric protein TRF1. TRF1, a component of the shelterin complex that protects the ends of our chromosomes, is modified by the PI3K/AKT signaling pathway, which senses the extracellular nutritional conditions. We generated *knock-in* human cell lines carrying mutant TRF1 variants unable to be modified by AKT. TRF1 mutant cells show decreased TRF1 binding to telomeres, increased DNA damage and accelerated telomere shortening. TRF1 mutant cells show an impaired TRF1 stability in response to proliferative extracellular signals and a decreased tumorigenesis potential, demonstrating that telomere function and telomere length are regulated by extracellular signals upstream of PI3K/AKT activation.

## Introduction

Telomeres are specialized structures at the chromosome ends, which are essential for chromosome-end protection and genomic stability [1]. Vertebrate telomeres consist of tandem repeats

Science and Innovation (SAF2017-82623-R (PM
and MAB)and SAF2015-72455-EXP (MAB)), the
Comunidad de Madrid Project (S2017/BMD-3770)
(MAB), the World Cancer Research (WCR) Project
(16-1177) (MAB), the European Research Council
(ERC-AvG Shelterines GA882385) (MAB) and the
Fundación Botín (Spain) (MAB). R.S-V is a
recipient of a doctoral scholarship from CONACYT-
México. The funders had no role in study design,
data collection and analysis, decision to publish, or
preparation of the manuscript.

**Competing interests:** The authors declare no
competing interests.

of the TTAGGG DNA sequence bound by the so-called shelterin complex, which encompasses
the TRF1, TRF2, TIN2, POT1, TPP1 and RAP1 proteins [2]. Shelterin ensures telomere pro-
tection by preventing end-to-end chromosome fusions, telomere fragility, and activation of
DNA repair activities at chromosome ends [3]. Some components of the shelterin complex
have also been described to regulate telomere length. In particular, TPP1 and POT1 have been
shown to regulate telomerase activity at telomeres [4,5]. In addition, expression of dominant-
negative mutants of TRF1 lead to telomere elongation [6], suggesting that TRF1 is a negative
regulator of telomere elongation by telomerase. However, the fact that *Trf1 knock-out* mice
and cells show a normal telomere length [7,8], indicates that the potential role for TRF1 in
telomere length regulation is still poorly understood.

In spite of these essential roles of shelterin in genome stability, telomere length regulation
and cellular viability, very little is known on the potential regulation of telomere protection by
key cell signaling pathways. We pioneered the understanding of the post-translational regula-
tion of shelterin components by using an unbiased chemical biology approach [9–11]. In par-
ticular, we showed that mouse TRF1 is post-translationally modulated by a number of key
cellular pathways, including the PI3K/AKT and Ras pathways [9–11]. In particular, we showed
that small molecule chemical inhibitors of these pathways impair proper telomere protection
[9–11]. Of particular interest is the regulation of TRF1 by the PI3K/AKT signaling pathway.
The PI3K/AKT pathway responds to different extracellular signals including immune regula-
tors, growth factors, and hormones [12,13]. In turn, activation of PI3K/AKT pathway regulates
a plethora of target proteins to control metabolism, cell proliferation, survival, and cell growth
[12,14]. We recently identified that an important target of the PI3K/AKT pathway is the TRF1
shelterin protein. In particular, we found that mouse TRF1 can be phosphorylated at three dif-
ferent residues by AKT and that these modifications are important for telomere protection [9].
However, a potential role for this TRF1 post-transcriptional modification *in vivo* is still
unknown.

Here, we set to address the *in vivo* relevance of the regulation of human TRF1 by the PI3K/
AKT pathway by generating *TRF1 knock-in* human cell lines carrying single mutations in the
AKT-dependent phosphorylation sites.

## Results

### Human TRF1 is phosphorylated by AKT at residues T273 and T358

In a previous report, we described that mouse TRF1 can be directly phosphorylated by AKT at
three independent residues (T248, T330, and S344), and that these phosphorylation events are
necessary for TRF1 stability [9]. Here, we first set to address whether these findings could be
extended to the human TRF1 protein. To this end, we carried out an *in vitro* kinase assay with
affinity purified wild-type human histidine-tagged-TRF1 (His-TRF1-WT) incubated with
human purified AKT in the presence of γ-32ATP (Materials and Methods). AKT yielded a
clear human TRF1 phosphorylation signal (p-His-TRF1 in **Fig 1A and 1B and S1 Source
Data**), which was dependent on the amount of purified GST-AKT used in the assay
(GST-AKT in **Fig 1A**), supporting the specificity of the signal. As a positive control, we also
detected a dose-dependent autophosphorylation of human AKT (**Fig 1A**).

Next, to identify the potential human TRF1 sites homologous to mouse AKT phosphoryla-
tion residues, we performed alignment of mouse and human TRF1 protein sequences using
the Basic Local Alignment Search Tool (BLAST) (NCBI) (**Fig 1C**). We identified human TRF1
residues T273, T344 and T358 as the homologous of mouse TRF1 residues T248, T330, and
S344, respectively, and as potential targets of AKT phosphorylation. Based on the consensus
sequence for AKT phosphorylation, RxRxxS/T [12,14], out of the three identified residues

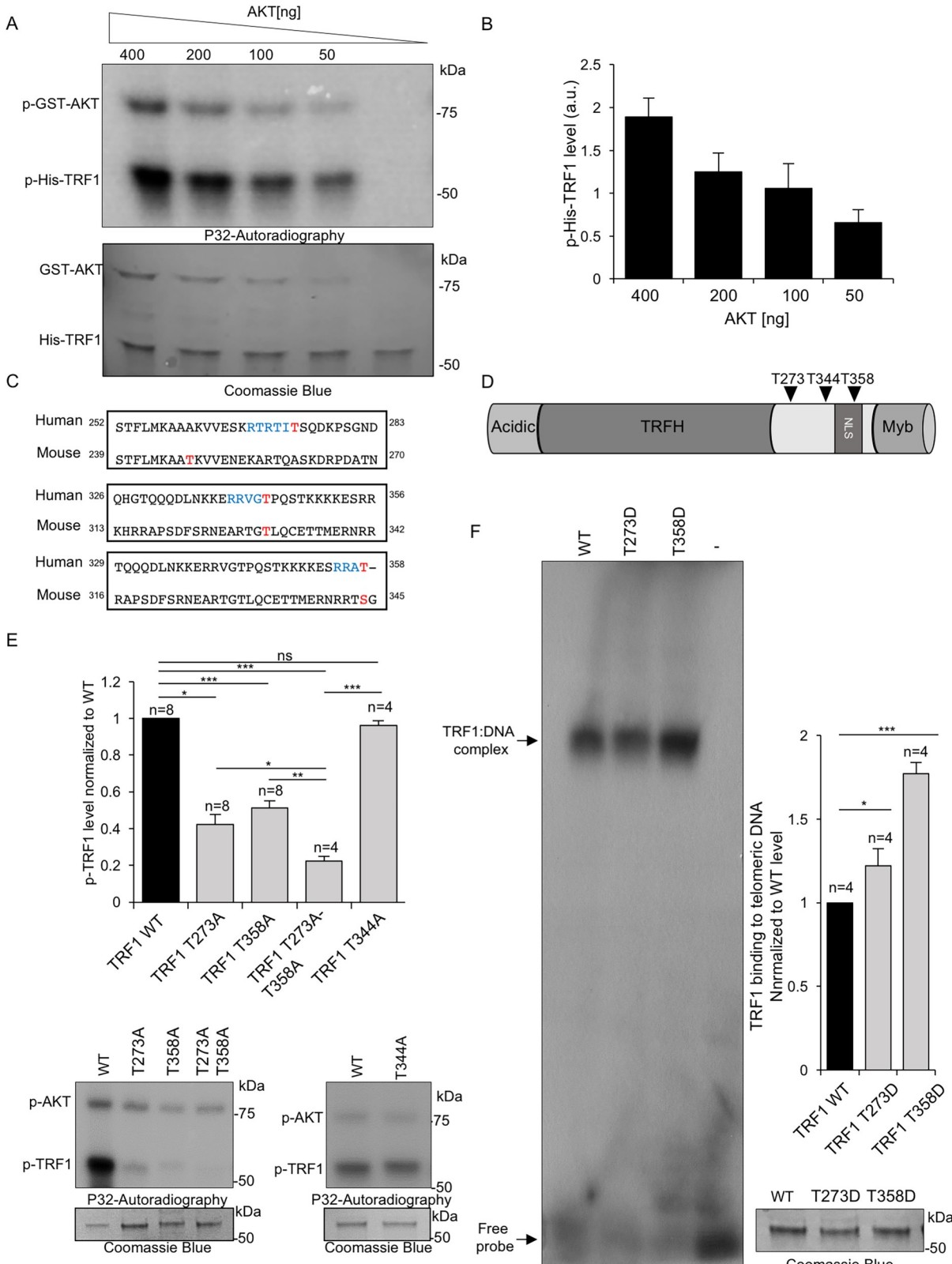

**Fig 1. Human TRF1 is phosphorylated *in vitro* by AKT. A.** Representative image of *in vitro* phosphorylation assay of purified His-TRF1-WT by GST-AKT1. Purified human wild-type His-TRF1 (1 μM) was incubated with different concentration of human GST-AKT1 in kinase buffer

containing 5 μCi [γ-32-P] ATP. The reaction mixtures were resolved by SDS-PAGE and subjected to autoradiography. The SDS-PAGE was Coomassie-stained to quantify total TRF1 as loading control (below panel). **B.** Quantification of phospho-TRF1 (p-TRF1) levels normalized to total TRF1. **C.** Alignment of mouse and human TRF1 protein sequences using The Basic Local Alignment Search Tool (BLAST) NCBI. Letters in blue indicated the AKT1 kinase consensus sequence and in red indicate the conserved threonine between human and mouse. **D.** Schematic representation of human TRF1 protein depicting the N-terminal acidic domain, the TRFH domain, the nuclear localization signal (NLS), and the Myb domain. The location of the threonine residues potentially phosphorylated by AKT (T273, T344, and T358) is indicated. **E.** Quantification of p-TRF1 levels of non-phosphorylatable TRF1 variants normalized to wild-type p-TRF1. Representative image of *in vitro* phosphorylation assay of purified His-TRF1 (WT), His-TRF1 (T273A), His-TRF1 (T344A), His-TRF1 (T358A) and His-TRF1 (T273A/T358A) by GST-AKT1 (0.2 μg). The reaction mixtures were resolved by SDS-PAGE and subjected to autoradiography. The SDS-PAGE was Coomassie-stained to quantify total TRF1 as loading control (below panel). **F.** Quantification of *in vitro* TRF1 binding to telomeric dsDNA $(TTAGGG)_7$ levels of His-tagged TRF1 WT and TRF1 phosphomimetic mutant variants (T273D and T358D). Representative image of the electrophoretic mobility shift assay (EMSA). The arrows show the position of the TRF1-DNA complexes and the free probe. A Coomassie-stained SDS-PAGE is shown as loading control (below panel). Error bars represent standard deviation. *n* number of independent experiments. Student's t test was used for statistical analysis, P values are shown. *, $p \leq 0.05$; **, $p \leq 0.01$; ***, $p \leq 0.001$; n.s. = not significant.

T273 constitutes the most likely candidate. Interestingly, all three potential TRF1 AKT phosphorylation sites were located near the nuclear localization signal domain (**Fig 1D**). To *in vitro* validate these potential phosphorylation sites we generated the His-tagged *TRF1* phospho-mutant alleles T273A, T334A, and T358A (Materials and Methods) (see **S1A and S1B Fig**). The affinity purified His-TRF1 wild-type or mutant proteins were incubated with human purified GST-AKT in the presence of γ-32ATP. We found significantly decreased TRF1 phosphorylation levels in T273A and T358A mutants but not in the T344A mutant compared to wild-type TRF1 (**Fig 1E** and **S1 Source Data**). To confirm that only T273 and T358 residues are important for AKT-dependent human TRF1 phosphorylation, we also assayed the phosphorylation level of the double T273A/ T358A mutant. The results show that AKT-mediated phosphorylation of the T273A/ T358A variant is fully blocked.

To address whether phosphorylation at residues T273 and T358 is important for TRF1 binding to telomeres, we generated phosphomimetic variants by replacing T273 and T358 residues for aspartic acid residues, i.e. TRF1-T273D and TRF1-T358D mutants. Affinity purified His-TRF1 wild-type or mutant proteins were incubated with radiolabeled double stranded telomeric DNA (an $TTAGGG_7$ oligo). The appearance of protein-DNA complexes by electrophoretic mobility shift assay (EMSA) showed that these phosphomimetic TRF1 mutant proteins bind more efficiently to telomeric DNA compared to wild-type TRF1 (**Fig 1F** and **S1 Source Data**). Together, these findings indicate that human TRF1 is phosphorylated by AKT at residues T273 and T358 and that these phosphorylation events are important for TRF1 binding to telomeric DNA *in vitro*.

## Generation of human *TRF1 knock-in* cell lines for AKT-dependent phosphorylation sites

In order to address the importance of AKT-dependent TRF1 phosphorylation *in vivo*, we set to generate human cell lines *knock-in* for *TRF1* alleles harboring inactivating mutations in AKT-dependent TRF1 phosphorylation sites. In particular, we used the CRISPR-Cas9 technology to generate non-phosphorylatable heterozygous and homozygous TRF1 mutant cell lines, namely *TRF1*$^{+/T273A}$, *TRF1*$^{T273A/T273A}$ and *TRF1*$^{+/T358A}$, *TRF1*$^{T358A/T358A}$. To this end, single-guide RNAs (sgRNA) showing the lower off-target mutation score were selected for each mutation using the online CRISPR [15]. In addition, to introduce the substitution of the corresponding bases of each amino acid, we designed a 100-nucleotide single-stranded DNA oligo (ssODN) as the homologous repair template. This ssODNs were designed to change the codon "ACT" by "GCT" and for codon-usage optimization as well as to introduce one additional base mutation to avoid re-cleaving and a new restriction site for subsequent clone genotyping (**Fig 2A**). The plasmid expressing EGFP, Cas9 and sgRNA were co-transfected with each

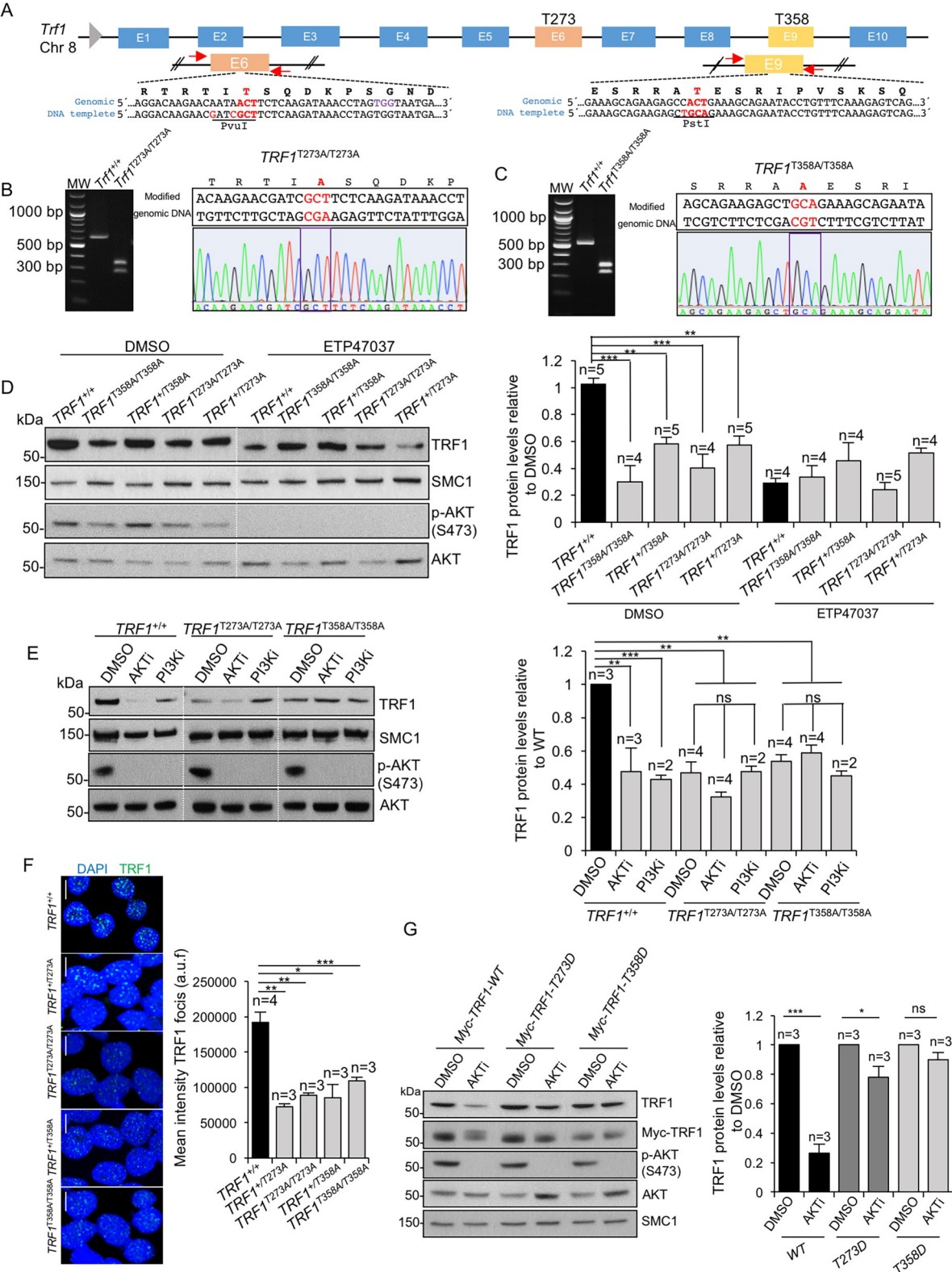

**Fig 2. Generation of human *TRF1 knock-in* cell lines. A.** Graphical scheme of CRISPR-Cas9-based strategy followed to substitute the ACT codon for GCT in exon 6 and 9 to generate *TRF1^T273A^* and *TRF1^T358A^* alleles, respectively. The genomic and the template DNA sequence are shown where the nucleotide substitution for amino acid change and for novel restriction site introduction are indicated in red. **B-C.** The

mutant $TRF1^{T273A}$ (A) and $TRF1^{T358A}$ (B) alleles were validated by RFLP analysis and Sanger sequencing. Representative images of ethidium bromide stained RFLP agarose gels (left panel) and of Sanger sequence spectrum (right panel) in $TRF1^{T273A/T273A}$ and $TRF1^{T358A/T358A}$ knock-in cells, respectively. **D.** Representative western blot images and quantification (right panel) of total nuclear of TRF1 protein levels in wild-type and heterozygous and homozygous mutant cells, treated either with DMSO or the PI3K inhibitor ETP47037 (10 µM). Phospho-AKT and total AKT were analyzed as positive control for the treatment. SMC1 was used as loading control. **E.** Representative western blot images and quantification (right panel) of total nuclear of TRF1 protein levels in wild-type and heterozygous and homozygous mutant cells, treated either with DMSO, AKT inhibitor MK22-06 (10 µM) or PI3K inhibitor ETP47037 (10 µM). Phospho-AKT and total AKT were analyzed as positive control for the treatments. SMC1 was used as loading control. **F.** Representative images and quantification (right) of TRF1 immunofluorescence in wild-type and mutant cells. Scale bars, 10 µm. **G.** Representative western blot images and quantification (right panel) of total nuclear of TRF1 protein levels in wild-type cells knocked down for endogenous TRF1 and overexpressing either wild-type MYC-tagged $TRF1$, MYC-tagged $TRF1^{T273D}$ or with MYC-tagged $TRF1^{T358D}$ alleles, treated either with DMSO or with AKT inhibitor MK22-06 (10 µM). Phospho-AKT and total AKT were analyzed as positive control for the treatment. SMC1 was used as loading control. Error bars represent standard deviation. $n$ number of independent experiments. Student's t test was used for statistical analysis, P values are shown. *, $p \leq 0.05$; **, $p \leq 0.01$; ***, $p \leq 0.001$; n.s. = not significant.

ssODN into Human embryonic kidney HEK 293T (ATCC CRL-3216). EGFP-positive cells were sorted by flow cytometry and amplified. The presence of the mutation in pooled cells was confirmed by PCR and restriction fragment length polymorphism analysis (RFLP). Single cell dilutions were plated onto 96-weel plates for clonal expansion and further RFLP identification of mutant clones [15]. The selected clones were then validated by Sanger DNA sequencing (**Fig 2B and 2C**). We isolated five independent heterozygous clones for both $TRF1^{+/T273A}$ and for $TRF1^{+/T358A}$ and two independent homozygous clones for $TRF1^{T273A/T273A}$ and for $TRF1^{T358A/T358A}$.

First, we performed Western blot analysis of nuclear TRF1 protein levels in wild-type and $TRF1$ heterozygous and homozygous mutant cell lines ($TRF1^{+/T273A}$, $TRF1^{T273A/T273A}$, $TRF1^{+/T358A}$, $TRF1^{T358A/T358A}$). We found that TRF1 protein levels were significantly lower in the $TRF1$ *knock-in* heterozygous cell lines than in wild-type cells, and we found a further decrease in TRF1 protein levels in the $TRF1$ *knock-in* homozygous cells (**Fig 2D and S1 Source Data**). To exclude clonal variations, we analyzed TRF1 protein levels in independent clonal isolates (**S2A and S2B Fig**). We did not find any significant differences between the different clones of the same genotype ruling out that reduced TRF1 levels were due to off-targets effects. These findings may suggest that the mutant TRF1 proteins are less stable than the wild-type TRF1 protein, in agreement with a role of AKT-dependent phosphorylation of these residues in TRF1 protein stability. Interestingly, the decrease in TRF1 protein levels induced by the $TRF1$ mutations was similar than that induced by treatment of $TRF1$ wild-type cells with an AKT chemical inhibitor (MK22-06) or with a chemical inhibitor against the AKT upstream kinase PI3K (ETP-47037) (**Fig 2D and 2E and S1 Source Data**), previously demonstrated by us to reduce TRF1 levels in mouse cells [9]. Of interest, treatment of the different $TRF1$ mutant cell lines with AKT or PI3K inhibitors did not further decrease TRF1 protein levels, demonstrating that the effect of AKT or PI3K inhibitors on TRF1 protein levels is mediated by the TRF1 residues identified in this manuscript (**Fig 2D and 2E**).

These results were confirmed by determining the nuclear TRF1 protein foci by immunofluorescence with anti-TRF1 antibodies [10,11] (**Fig 2F**). In particular, we found that both $TRF1$ heterozygous and homozygous mutant cell lines ($TRF1^{+/T273A}$, $TRF1^{T273A/T273A}$, $TRF1^{+/T358A}$, $TRF1^{T358A/T358A}$) showed significantly decreased TRF1 protein foci fluorescence compared to the TRF1 wild-type controls (**Fig 2F**).

Wild type cells were knocked down for the endogenous TRF1 by using a sh-$TRF1$ (**S2C Fig and S1 Source Data**) and transfected with a vector harboring either wild-type MYC-tagged $TRF1$, MYC-tagged $TRF1^{T273D}$ or with MYC-tagged $TRF1^{T358D}$ alleles. Cells overexpressing wild type and phosphomimetic $TRF1$ alleles were then treated either with DMSO or with AKT chemical inhibitor (MK22-06) for 24 hours. Western blot analysis of nuclear TRF1 protein

levels were performed using anti-TRF1 or anti-MYC antibodies. The results showed that the levels of wild type MYC-TRF1 decreased to 70% upon AKTi treatment while the levels of both MYC-TRF1[T273D] and MYC- TRF1[T358D] remained largely unaffected (**Fig 2G and S1 Source Data**). These results further demonstrate that residues T273 and T358 are key for AKT-dependent TRF1 stability.

## Human *TRF1 knock-in* cell lines for AKT-dependent phosphorylation sites show increased proteasome-mediated TRF1 protein degradation

Next, we set to determine the molecular mechanisms underlying decreased TRF1 protein levels in the *TRF1* mutant cells. Of relevance, we could not attribute the changes in TRF1 protein expression to changes in *TRF1* mRNA expression. In particular, although we detected significantly altered *TRF1* mRNA levels in cells carrying heterozygous *TRF1* alleles, this effect was not detected in cells carrying the *TRF1* homozygous alleles (**Fig 3A and S1 Source Data**).

To directly study TRF1 protein stability, we performed a cycloheximide chase assay of protein degradation (Materials and Methods). Cells carrying the mutant *TRF1* alleles showed significantly lower TRF1 protein levels at different times after cycloheximide treatment compared to wild-type cells (**Fig 3B and S1 Source Data**), suggesting a decreased TRF1 mutant protein stability. Interestingly, treatment with the proteasome inhibitor bortezomib rescued TRF1 protein levels in the different *TRF1* mutant cell lines, indicating that lower protein stability of the TRF1 mutants is caused by proteasome-mediated degradation (**Fig 3B**). As a negative control, we also analyzed the stability of a different shelterin component, RAP1 in both wild-type and *TRF1* mutant cell lines (**Fig 3C and S1 Source Data**). We did not find differences in RAP1 stability between wild-type and *TRF1* mutant cell lines (**Fig 3C**), indicating that the AKT-dependent TRF1-dependent regulation does not affect RAP1 protein levels. Interestingly, it is known that the inhibition of protein synthesis by cycloheximide induces phosphorylation/activation of AKT [16]. Thus, we set to address AKT phosphorylation after cycloheximide treatment in the different *Trf1* wild-type and mutant cell lines. We found increased AKT phosphorylation at Ser473 both in *TRF1* will-type and mutant cells (**Fig 3D and S1 Source Data**). We confirmed these results by measuring phosphorylation of an AKT downstream substrate, the ribosomal protein S6, and again found increased levels of p-S6 in both *TRF1* will-type and mutants cell lines, in agreement with increased phospho-AKT (**Fig 3E and S1 Source Data**). These findings indicate that the cycloheximide-dependent AKT phosphorylation stabilizes the wild-type TRF1 protein, but this does not occur in the *TRF1* mutant cell lines, demonstrating the importance of T273 and T358 TRF1 residues in AKT-mediated TRF1 post-translational regulation.

## Human *TRF1 knock-in* cell lines for AKT-dependent phosphorylation sites show increased global and telomeric DNA damage and increased telomere fragility

TRF1 has an essential role in protecting the chromosome ends from eliciting a persistent DNA damage response (DDR), as well as in preventing telomere fragility as detected by the occurrence of multiple telomere signals per chromosome end (MTS) [7,8]. Thus, we first set to address the effects of mutant TRF1 proteins in both global and telomeric DNA damage. To this end, we performed either immunofluorescence with γH2AX to detect global DNA damage or double immunofluorescence of γH2AX and the telomeric protein RAP1 to detect DNA damage specifically located at telomeres (the so-called telomere induced foci or TIFs) [17]. We found that homozygous *TRF1* mutant cell lines showed significantly increased levels of both global and telomeric DNA damage while heterozygous *TRF1* mutant cell lines showed normal

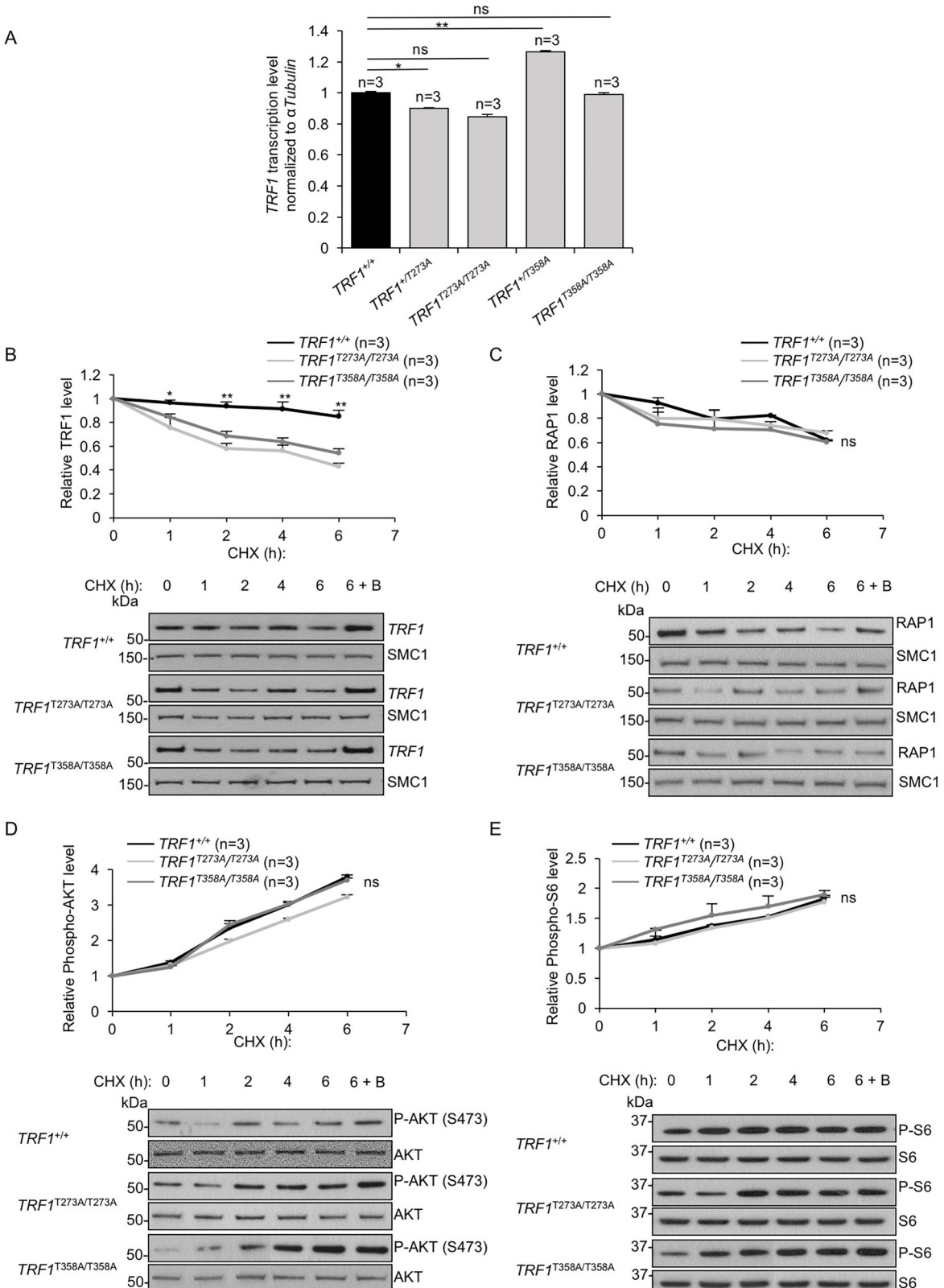

**Fig 3. AKT-mediated TRF1 phosphorylation regulates TRF1 protein stability. A.** Relative *TRF1* mRNA expression level in wild-type and in mutant cell lines. **B-C.** TRF1 wild type and mutant variants (B) and RAP1 (C) turnover was determined in wild type, *TRF1^T273A/*

$^{T273A}$ and *TRF1*$^{T358AT/358A}$ cell lines by cycloheximide (CHX) pulse chase. Cells were treated with 100 μg/ml of CHX in the absence or presence of 5 μM bortezomib. At the indicated times, the cells were harvested to prepare nuclear extracts, followed by Western blot analysis against TRF1 (B) or RAP1 (C) and SMC1 (as loading control) (lower panels). **D-E.** Phospho-AKT (D) and phospho-S6 (E) levels were determined in wild type, *TRF1*$^{T273A/T273A}$ and *TRF1*$^{T358AT/358A}$ cell lines at the indicated times during CHX pulse chase. Total AKT (D) and S6 (E) was used as loading control. The graph represents the mean value for each time point and corresponding standard deviation. *n* number of independent experiments. Student's t test was used for statistical analysis, P values are shown. $^{*}$, p ≤ 0.05; $^{**}$, p ≤ 0.01; n.s. = not significant.

global and telomeric DNA damage levels (**Fig 4A and 4B and S1 Source Data**), indicating that AKT-dependent TRF1 phosphorylation is essential for preventing telomeres from eliciting a DNA damage response. We checked the cell cycle status of wild-type and mutant cells and found no significant differences among the genotypes, ruling out cell cycle-dependent variations in the DDR blot (**S3A Fig and S1 Source Data**).

TRF1 inhibition has been previously shown to induce the so-called multitelomeric signals (MTS), which are associated to increased telomere fragility and increased telomere damage [7,8]. Thus, we next tested whether the non-phosphorylatable *TRF1* mutant cells lines showed an increase in MTSs. To this end, we performed telomere quantitative fluorescence *in situ* hybridization (Q-FISH) on metaphase spreads to visualize telomeres (Materials and Methods) in *TRF1* wild-type, as well as *TRF1* heterozygous and homozygous mutant cell lines (**Fig 4C and S1 Source Data**). We found that homozygous *TRF1* mutant cells showed significantly elevated number of MTSs compared to the *TRF1* wild-type controls. Of interest, one of the heterozygous mutants, *TRF1*$^{+/T358A}$, also showed significantly increased MTSs in this assay, suggesting haploinsufficiency for the MTS phenotype.

TIF and MTS induction after TRF1 depletion is accompanied by activation of the ATM/ATR signaling pathways and the phosphorylation of their downstream checkpoint kinases CHK2 and CHK1 [7,8]. To address the impact of TRF1-T273A and TRF1-T358A mutant variants on DDR activation, we analyzed phospho-CHK2 and phospho-CHK1 levels by Western blot (**S3B and S3C Fig and S1 Source Data**). The results show that mutant cells present increased phospho-CHK2 and phospho-CHK1 levels compared to wild-type cells, a DDR characteristic of replicative damage.

## *TRF1 knock-in* cell lines for AKT-dependent phosphorylation show reduced proliferation ability

The PI3K/AKT pathway is known to promote cell proliferation and survival throughout modulation of different targets [14,18]. As we have shown here that TRF1 is a *bona fide* target of this pathway, we set to address whether the human *TRF1 knock-in* cell lines expressing non-phosphorylatable TRF1 proteins in the AKT phosphorylation sites, showed any cell proliferation defects. Interestingly, both heterozygous and homozygous *TRF1* mutant cell lines showed reduced cell proliferation as compared to the wild-type *TRF1* controls (**S4 Fig and S1 Source Data**). The *TRF1*$^{+/T273A}$ heterozygous cell line showed a severe phenotype which was comparable to that of the homozygous cell lines (**S4 Fig**). These results suggest that post-translational modifications in the shelterin complex at chromosome ends can modulate the ability of cells to proliferate.

## *TRF1 knock-in* cell lines for AKT-dependent phosphorylation show telomere length defects

Telomere length regulation is a key process in cancer and aging. On one hand, telomere shortening associated to cell division and tissue repair throughout the organismal lifespan is

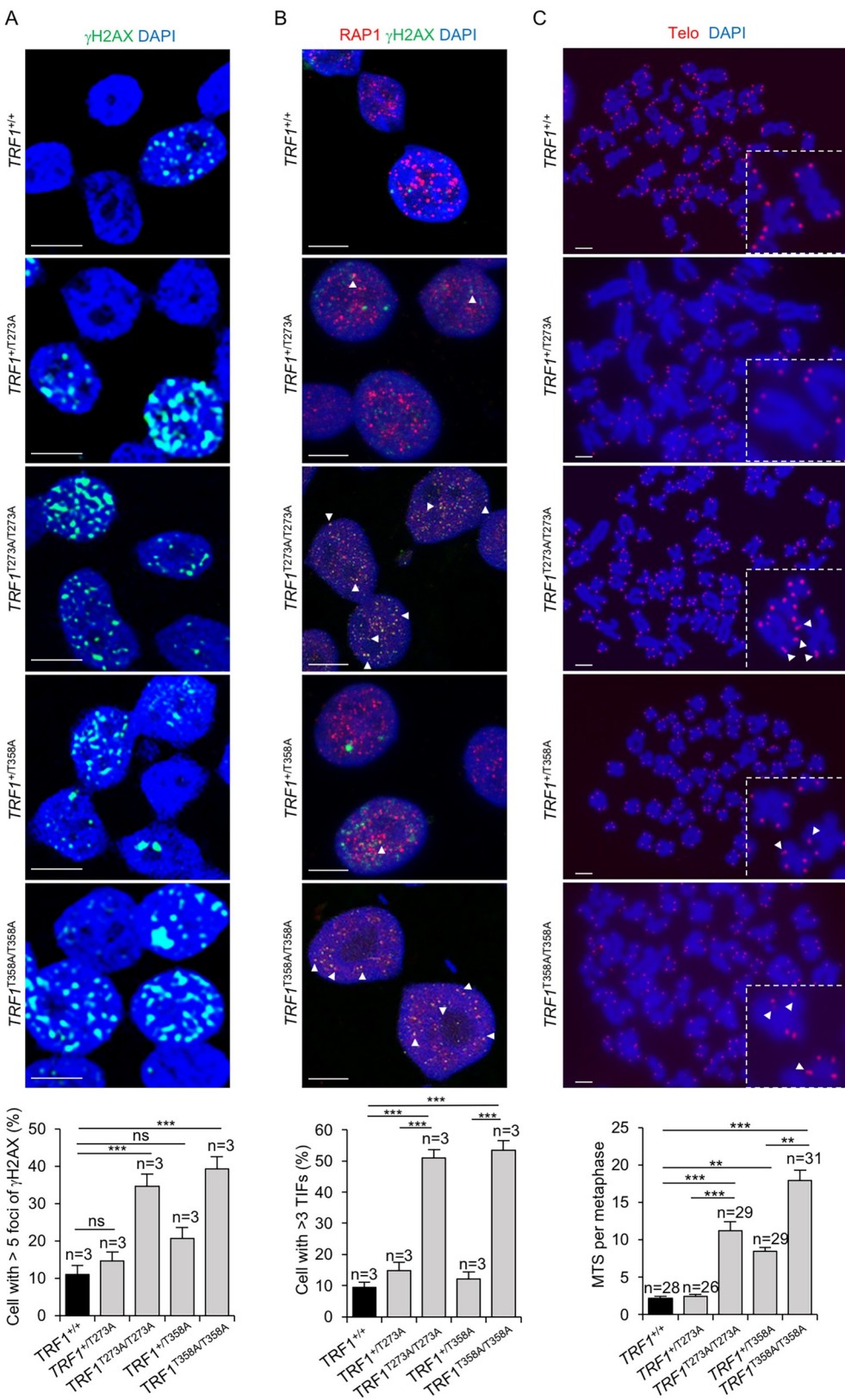

**Fig 4. AKT-dependent TRF1 phosphorylation is required for telomere protection. A.** Representative images (above panel) and quantification (below panel) of percentage of γH2AX positive cells in wild type, *TRF1⁺/ᵀ²⁷³ᴬ, TRF1ᵀ²⁷³ᴬ/*

$T273A$, $TRF1^{+/T358A}$ and $TRF1^{T358A/T358A}$ cell lines. Scale bars, 10 μm. **B.** Representative images (above panel) and quantification (below panel) of percentage of cells presenting three or more γH2AX and RAP1 co-localizing foci (TIFs) (white arrowhead). Scale bars, 10 μm. **C.** Representative images (above panel) and quantification (below panel) of multitelomeric signals (MTS) (white arrow heads). Insets represent high magnification images. Scale bars, 5μm. Error bars represent standard deviation. *n* number of independent experiments (A,B) or number of metaphases (C). Student's t test was used for statistical analysis, P values are shown. *, p ≤ 0.05; **, p ≤ 0.01; ***, p ≤ 0.001.

proposed to be one of the molecular causes underlying aging and age-related diseases [19]. On the other hand, cancer cells need maintenance of telomeric repeats in order to divide indefinitely and they achieve that by activating telomere maintenance mechanisms such as telomerase activity in the majority of tumors, or less frequently, activating alternative mechanisms of telomere lengthening based on homologous recombination between telomeric sequences (Alternative Lengthening of Telomeres or ALT) [20]. As some reports suggested a role for TRF1 as a negative regulator of telomere length [6] while other reports did not show any telomere length defects upon TRF1 genetic deletion in mice [7,8], here we set to investigate the impact of AKT-dependent phosphorylation of TRF1 in telomere length regulation *in vivo*. To this end, we first studied telomere length by using telomere restriction fragment analysis (TRF) based on Southern blotting [4,21](Materials and Methods). We found significantly decreased telomere length both in heterozygous and homozygous *TRF1* mutant cell lines compared to wild-type cells (**Fig 5A**). These defects were not aggravated when treating with ETP47034, an PI3Kα chemical inhibitor known to inhibit AKT phosphorylation (**Fig 5A**), again suggesting that the effects of the inhibitor are mediated by the AKT-dependent phosphorylation of TRF1. We confirmed shorter telomeres in cells carrying *TRF1* homozygous mutant alleles by an independent technique based on telomere fluorescence known as quantitative telomere FISH or Q-FISH (Materials and Methods), which measures individual telomere signals (**Fig 5B** and **S1 Source Data**) [21,22]. In agreement with telomere maintenance defects in the *TRF1* mutant alleles, we found that homozygous *TRF1* mutants showed significantly shorter telomeres than TRF1 wild-type cells as well as a dramatic increase in the percentage of short telomeres (defined as the % of telomeres showing a lower fluorescence intensity than the 15% percentile of control wild-type cells) (**Fig 5B**). Furthermore, we also found and increase in the number of undetectable telomere signals (referred here as "signal-free" ends) per metaphase in *TRF1* homozygous mutant cells (**Fig 5B**), again indicative of impaired telomere maintenance. The shorter telomere phenotype was confirmed in independent clones, ruling out potential clonal variations and demonstrating that TRF1 mutant variants are causative of telomere length defects (**S5 Fig**).

In agreement with defective telomere maintenance of AKT non-phosphorylatable TRF1 mutants, we observed that the shorter telomere phenotype was aggravated with increasing cell passage in both homozygous mutant cell lines compared to wild-type cells (**Fig 5C**). Importantly, the short telomere phenotype as well as the presence of short telomeres and of "signal-free ends" was rescued upon over-expression of wild-type flag-tagged TRF1, but only in $TRF1^{T273A/T273A}$ cells and not in $TRF1^{T358A/T358A}$ cells (**Figs S6A** and **5B and 5C,**). These findings indicate that telomere shortening phenotype induced by the $TRF1^{T273A/T273A}$ mutant protein is the direct effect of the mutation as this effect can be rescued by over-expression of the wild-type TRF1 protein. However, telomere shortening induced by the $TRF1^{T358A/T358A}$ mutant, is likely to be indirect as it cannot be rescued by over-expression of the wild-type TRF1 protein. It is possible that the $TRF1^{T358A/T358A}$ mutation may also be modifying the structure of the telomere capping complex, thus impeding proper access of the wild-type TRF1 protein.

To understand the consequences of TRF1 abrogation, we downregulated TRF1 protein levels by using and shRNA against TRF1 (Materials and Methods). To this end, wild-type and

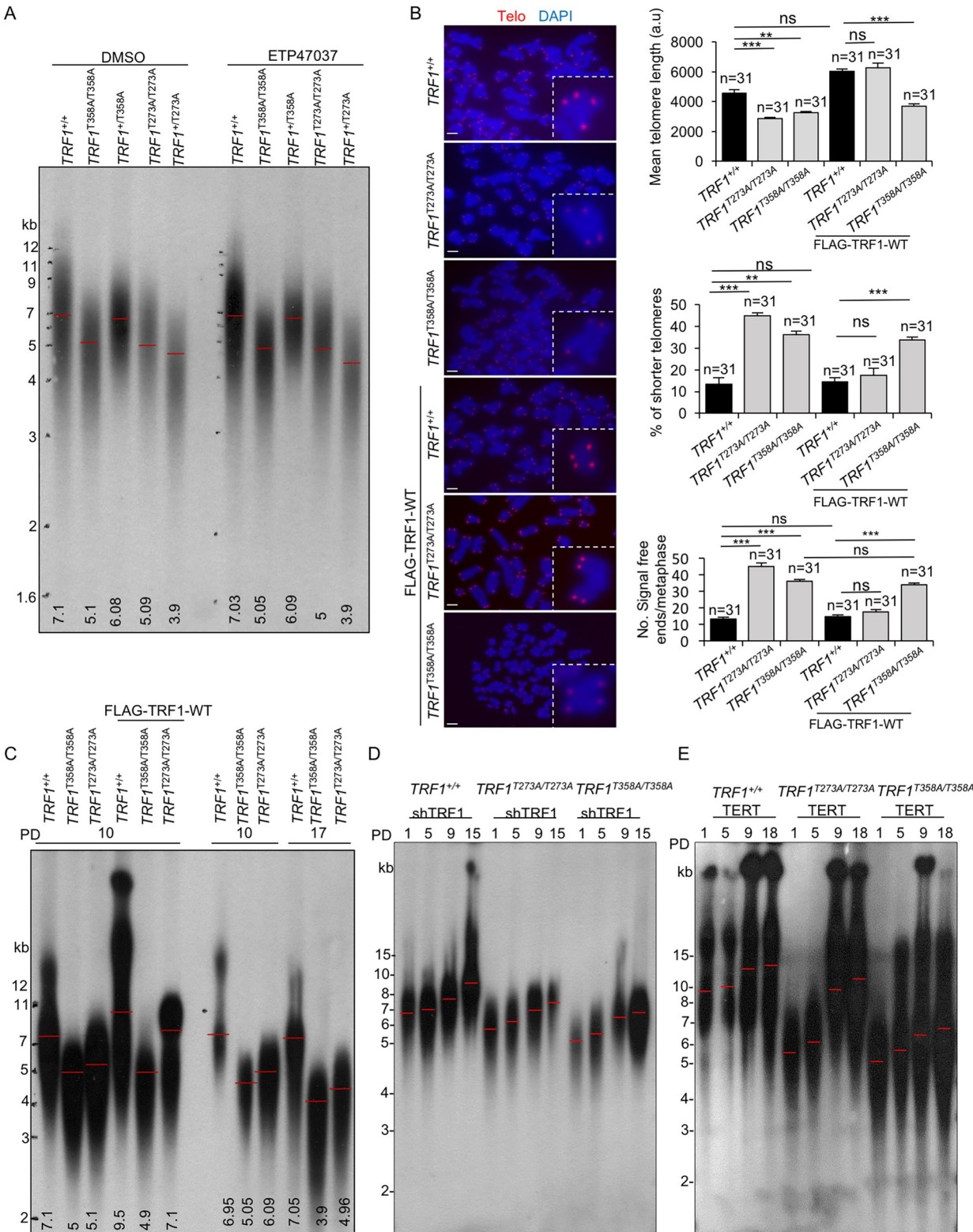

**Fig 5. AKT-dependent TRF1 phosphorylation is required for telomere length maintenance. A.** Representative image of telomeric restriction fragment (TRF) blot of *TRF1^+/+^*, *TRF1^+/T273A^*, *TRF1^T273A/T273A^*, *TRF1^+/T358A^* and *TRF1^T358A/T358A^* cell lines. Numbers refer to molecular weight standards in Kb. Mean telomere length for each cell line is shown at the base of the lanes. **B.** Representative Q-FISH images of metaphases spreads from *TRF1^+/+^*, *TRF1^T273A/T273A^* and *TRF1^T358A/T358A^* cell lines either non-transfected or expressing FLAG-TRF1-WT. Insets represent high

magnification images. Scale bars, 5μm. Quantification of the mean telomere length, the percentage of short telomeres and the number of signal-free ends per metaphase by Q-FISH analysis. The percentage of short telomeres is defined as fluorescence intensity less that the 15th percentile of the fluorescence intensity values of the wild type control cells. n, number of metaphases. **C.** TRF blots of $TRF1^{+/+}$, $TRF1^{T358A/T358A}$ and $TRF1^{T273A/T273A}$ cell lines at the indicated population doubling (PD). Lanes 4–6, $TRF1^{+/+}$, $TRF1^{T358A/T358A}$ and $TRF1^{T273A/T273A}$ cells were transfected with FLAG-TRF1-WT. Mean telomere length for each cell line is shown at the base of the lanes. **D.** TRF blots of $TRF1^{+/+}$, $TRF1^{T358A/T358A}$ and $TRF1^{T273A/T273A}$ cells transfected with a $pLKO$-$shTRF1$ at the indicated population doubling (PD). **E.** TRF blots of $TRF1^{+/+}$, $TRF1^{T358A/T358A}$ and $TRF1^{T273A/T273A}$ cells transfected with $pBabe$-$TERT$ at the indicated population doubling (PD).

mutant cell lines were transfected with an shTRF1 vector to downregulate TRF1 expression. We confirmed decreased TRF1 mRNA and protein levels in the cell lines transduced with the shTRF1 (S6B and S6D Fig and S1 Source Data). We next measured telomere length by telomere restriction analysis at increasingly cell passages (Fig 5D). In agreement with previous reports [6], TRF1 knock-down induced a significant progressive telomere lengthening in both wild-type and in TRF1 mutant cells confirming a TRF1 role as a negative regulator of telomere elongation. We confirmed these results by measuring telomere length by Q-FISH in metaphase spreads from $TRF1^{+/+}$, $TRF1^{T273A/T273A}$ and $TRF1^{T358A/T358A}$ cells expressing shTRF1 at passage 14 (S6D Fig). These results also indicate that telomere length defects shown by $TRF1^{T273A/T273A}$ and $TRF1^{T358A/T358A}$ mutant cells are not due to the lower protein stability and consequent reduction in TRF1 protein levels in mutant cells (Figs 2D, 2F, and 3B) but rather to a *de novo* gain of function of these mutations (Fig 5D).

The progressive telomere elongation observed upon TRF1 downregulation in wild-type, $TRF1^{T273A/T273A}$ and $TRF1^{T358A/T358A}$ cells indicates a telomerase-dependent telomere lengthening and suggests that depleted TRF1 levels may allow telomerase to access telomeres. In order to address whether telomerase overexpression was able to bypass defective telomere maintenance in the mutant TRF1 cell lines, we transfect wild-type, $TRF1^{T273A/T273A}$ and $TRF1^{T358A/T358A}$ cells with a vector containing the human TERT [23]. We confirmed telomerase over expression by qPCR (S6E Fig and S1 Source Data). We next measured telomere length by telomere restriction analysis at increasingly cell passages (Fig 5E). The results show that telomerase overexpression induces a significant and progressive telomere lengthening in both wild-type and in TRF1 mutant cells.

We and others previously showed that induction of pluripotency by transduction of the Yamanaka factors in mouse and human *induced pluripotent stem* (iPS) cells leads to *de novo* telomere elongation mediated by telomerase [24,25]. Thus, we next set to address telomere length dynamics upon induction of pluripotency in $TRF1^{+/+}$ cells, as well as in the $TRF1^{T273A/T273A}$ and $TRF1^{T358A/T358A}$ mutants. To this end, wild-type and mutant cell lines were transfected with OCT4, SOX2, KLF4 and c-MYC containing vectors to induce reprogramming. First, in agreement with induction of pluripotency, we observed increased Nanog mRNA levels in all the cell lines upon reprogramming with the Yamanaka factors (S7A Fig and S1 Source Data). Of interest, TRF1 transcriptional levels did not change upon induction of reprogramming (S7B Fig and S1 Source Data), in contrast with previous reports in which TRF1 was found to be over-expressed upon reprogramming of mouse embryonic fibroblasts [26]. This could be due to the fact that the HEK293T cell line used here is embryo-derived and thus have already elevated TRF1 levels. We next measured in parallel telomere length both by telomere restriction analysis and by Q-FISH in metaphase spreads from reprogrammed $TRF1^{+/+}$, $TRF1^{T273A/T273A}$ and $TRF1^{T358A/T358A}$ cells at passage 14 (S7C and S7D Fig and S1 Source Data). Although $TRF1^{T273A/T273A}$ and $TRF1^{T358A/T358A}$ cells showed shorter telomeres than wild-type controls before reprogramming, induction of pluripotency resulted in significant telomere elongation both in TRF1 wild-type and TRF1 mutant cells compared to the non-reprogrammed counterparts (S7C and S7D Fig). As the levels of TRF1 did not significantly

change during reprogramming (**S7B Fig**), the observed telomere elongation suggests that telomerase upregulation upon induction of pluripotency [24] was able to bypass defective telomere maintenance in the mutant TRF1 cell lines.

## Shelterin complex formation in *TRF1 knock-in* cell lines for AKT-dependent phosphorylation

To address whether the expression of different proteins of the shelterin complex was normal in the $TRF1^{T273A/T273A}$ and $TRF1^{T358A/T358A}$ knock-in cell lines, we first analyzed the abundance of different shelterins by performing Western blotting of whole cell lysates of wild-type, $TRF1^{T273A/T273A}$ and $TRF1^{T358A/T358A}$ cells, as well as the same cells overexpressing wild-type FLAG-TRF1 (**Fig 6A and 6B** and **S1 Source Data**). As expected, we found a decrease of TRF1 protein levels in the $TRF1^{T273A/T273A}$ and $TRF1^{T358A/T358A}$ cells and this was rescued when expressing the wild-type FLAG-TRF1 in both mutants (**Fig 6A and 6B**). In spite of decreased TRF1 levels, we observed normal levels of the rest of the shelterin components in $TRF1^{T273A/T273A}$ and $TRF1^{T358A/T358A}$ cells (**Fig 6A and 6B**), suggesting that the mutant TRF1 proteins do not interfere with the stability of shelterin components.

To address whether the non-phosphorylatable TRF1-T273A and TRF1-T358A mutants were affected in binding to known TRF1-interacting shelterin components, namely POT1 and TIN2 [27–29], we transfected wild-type cells with either Flag-TRF1, Flag-TRF1-T273A or Flag-TRF1-T358A in HEK293T cells and performed co-immunoprecipitation with anti-Flag antibody (**Fig 6C and 6D** and **S1 Source Data**). As control, expression of the different TRF1 mutant proteins was indistinguishable from that of wild-type TRF1 (**Fig 6C**). Interestingly, similar amounts of the POT1 and TIN2 shelterins were pulled down in cells expressing either Flag-TRF1, Flag-TRF1-T273A or Flag-TRF1-T358A, indicating that these mutations do not interfere with TRF1 interactions with the shelterin components (**Fig 6D** and **S1 Source Data**). Finally, by performing chromatin immunoprecipitation experiments in these transfected wild-type cells, we observed a 60% and 40% reduced TRF1-T273A and TRF1-T358A binding to telomeric DNA *in vivo*, respectively (**Fig 6E** and **S1 Source Data**). Interestingly, binding of TRF2 to the telomere was unaffected in spite of the decreased TRF1 levels bound to telomeres (**Fig 6F** and **S1 Source Data**). These findings are in agreement with normal TRF2 foci formation in the cells genetically depleted for TRF1 [8], suggesting that the AKT-dependent TRF1 phosphorylation specifically regulates TRF1 binding to telomeres and that this is sufficient to affect telomere protection and telomere length even in the presence of normal TRF2 levels.

## Extracellular cues are transduced to telomeres through PI3K/AKT signaling pathway

The PI3K/AKT pathway responds to different extracellular signals to regulate cell proliferation and growth [12,14]. TRF1 is required for proper telomeric DNA replication [7,8]. However, a direct cause-effect connection between extracellular signaling and changes in telomere regulation have not been demonstrated to date. Since we found here that human TRF1 is regulated by PI3K/AKT pathway, we next set to address the potential *in vivo* role of AKT-dependent TRF1 phosphorylation in transducing extracellular signaling to telomeres. To this end, we analyzed TRF1 protein levels under normal fed conditions in DMEM+FBS, starved for 24 hours in DMEM without FBS, and either refed for 5 hours with FBS or with FBS plus PI3K inhibitor (ETP47037) in *TRF1* wild-type and homozygous *knock-in* cells for non-phosphorylateble *TRF1* alleles ($TRF1^{T358A}$ and $TRF1^{T273A}$). Interestingly, we find that in wild-type cells, TRF1 levels decrease by 50% under starvation conditions and that upon FBS administration TRF1 levels increase to initial levels. The increase in TRF1 levels upon re-feeding was fully blocked

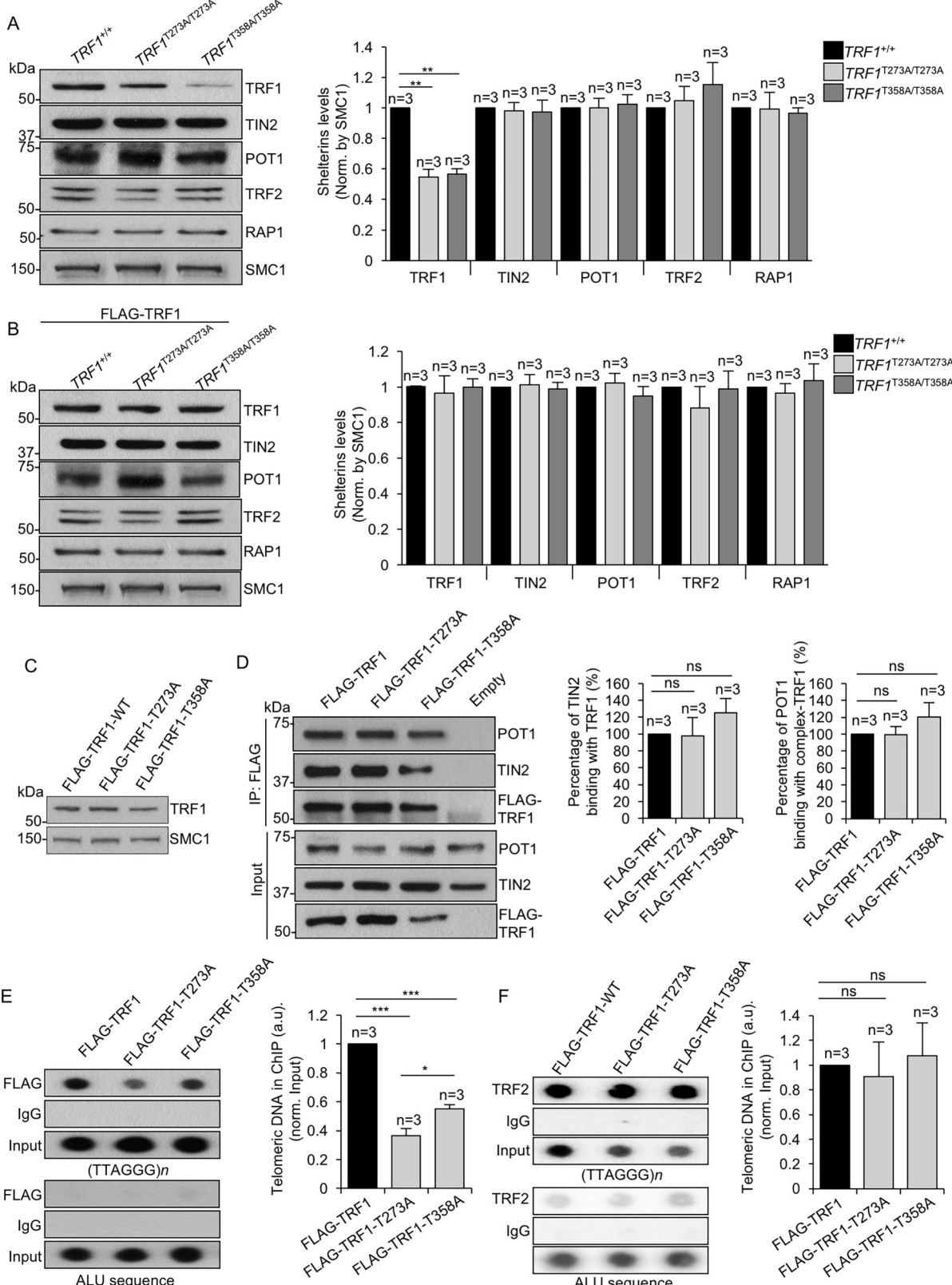

**Fig 6. AKT-dependent TRF1 phosphorylation is required for proper telomere binding but dispensable for the interaction with other shelterins. A.** Representative western blot images (left) and quantification (right) of TRF1, TIN2, POT1, TRF2 and RAP1 protein levels in

*TRF1$^{+/+}$*, *TRF1$^{T358A/T358A}$* and *TRF1$^{T273A/T273A}$* cells. **B.** Representative western blot images (left) and quantification (right) of TRF1, TIN2, POT1, TRF2 and RAP1 protein levels in *TRF1$^{+/+}$*, *TRF1$^{T358A/T358A}$* and *TRF1$^{T273A/T273A}$* cells expressing FLAG-TRF1-WT. **C.** Total TRF1 levels in wild type HEK293T cells expressing either Flag-TRF1, Flag-TRF1-T273A or Flag-TRF1-T358A by western blot. **D.** Flag-TRF1, Flag-TRF1-T273A and Flag-TRF1-T358A co-immunoprecipitate with endogenous TIN2 and POT1. Co-immunoprecipitation was performed from lysates of HEK293T cells expressing either the empty vector or the indicated Flag-*TRF1* alleles. Quantification of TIN2 and POT1 pulled down with anti-Flag. **E.-F.** Representative images of chromatin immunoprecipitation (ChIP) of telomeric DNA and of ALU sequences with anti-Flag (E) and with anti-TRF2 (F) of HEK293T cells expressing the indicated Flag-*TRF1* alleles. DNA input signal is also shown. Quantification of telomeric DNA pulled down with anti-Flag (E) or with anti-TRF2 (F) is shown to the right. ChIP values are normalized by the input of each individual sample. Bars and error bars represent mean values ± SE, n = number of independent experiments. Student's t-test was used for the statistical analysis; $^{*}$p < 0.05, $^{**}$p < 0.01, $^{***}$p < 0.001 ns, no significant.

in the presence of the PI3Ki (**Fig 7A** and **S1 Source Data**), indicating that is dependent on the PI3K/AKT pathway. To further demonstrate this, we repeated the experiment in the different TRF1 mutant cells impaired for AKT-dependent TRF1 phosphorylation. Interestingly, in the mutant cell lines, the levels of TRF1-T273A and TRF1-T358A proteins remained unaltered upon changes in the extracellular nutrient availability. As control for PI3K activation/inactivation we assayed phospho-AKT (S473) levels (**Fig 7A**).

As the PI3K/AKT pathway is known to be a major signaling pathway downstream of insulin signaling, we next addressed whether addition of insulin to starved cells (DMEM without FBS) affected TRF1 levels (**Fig 7B** and **S1 Source Data**). Addition of insulin to starved wild-type cells induced a 80% increase in TRF1 levels, which was impaired in the presence of either PI3K (ETP47037) or AKT chemical inhibitors (MK22-06) (**Fig 7B**). Again, when we performed the same experiment in different TRF1 mutant cells impaired for AKT-dependent TRF1 phosphorylation, the levels of non-phosphorylatable TRF1 variants, TRF1-T273A and TRF1-T358A, remained unchanged upon addition of either insulin or insulin plus AKTi/PI3Ki. To address whether the levels of other shelterin components did also change upon extracellular insulin signals, we checked TIN2, TRF2 and RAP1 protein levels in these conditions. TIN2, TRF2 and RAP1 levels remained unaltered upon addition of either insulin or insulin plus AKTi/PI3Ki (**Fig 7B**). As control for PI3K activation/inactivation we assayed phospho-AKT (S473) levels (**Fig 7B**).

In summary, these results demonstrate the importance of AKT-dependent TRF1 phosphorylation in residues T273 and T358, for the proper transduction of extracellular cues to telomeres.

## AKT-dependent TRF1 phosphorylation impacts on the tumorigenic capability of cancer cells

Both the PI3K/AKT and the telomere maintenance pathways are frequently mutated in cancer [20,30]. Here, we set to address whether human cancer cells mutant for the AKT-dependent TRF1 phosphorylation sites identified here retained their tumorigenic ability. To this end, HEK 293T *TRF1* wild-type and homozygous *knock-in* cells for non-phosphorylatable *TRF1* alleles (*TRF1$^{T358A}$* and *TRF1$^{T273A}$*), were subcutaneously injected into nude mice (Material and Methods). Tumor growth was measured by caliper every second day. Xenografts derived from *TRF1$^{T358A/T358A}$* and *TRF1$^{T273A/T273A}$* cells showed a delayed tumor onset and a significant slower tumor growth as compared to *TRF1$^{+/+}$* cells (**Fig 8A** and **S1 Source Data**). All the mice were sacrificed 15 days post-injection when wild-type tumors reached large volume (1500mm$^{3}$). Postmortem tumor analysis of mutant xenografts (*TRF1$^{T358A/T358A}$* and *TRF1$^{T273A/T273A}$*) revealed a significant reduction in tumor size and weight as compared to wild-type tumors (**Fig 8A and 8B** and **S1 Source Data**). Tumors derived from *TRF1$^{T358A/T358A}$* and *TRF1$^{T273A/T273A}$* cells showed a significant decrease in the proliferation marker Ki67 and an increase in the DNA damage marker γH2AX (**Fig 8C, 8D, and 8E** and **S1 Source**

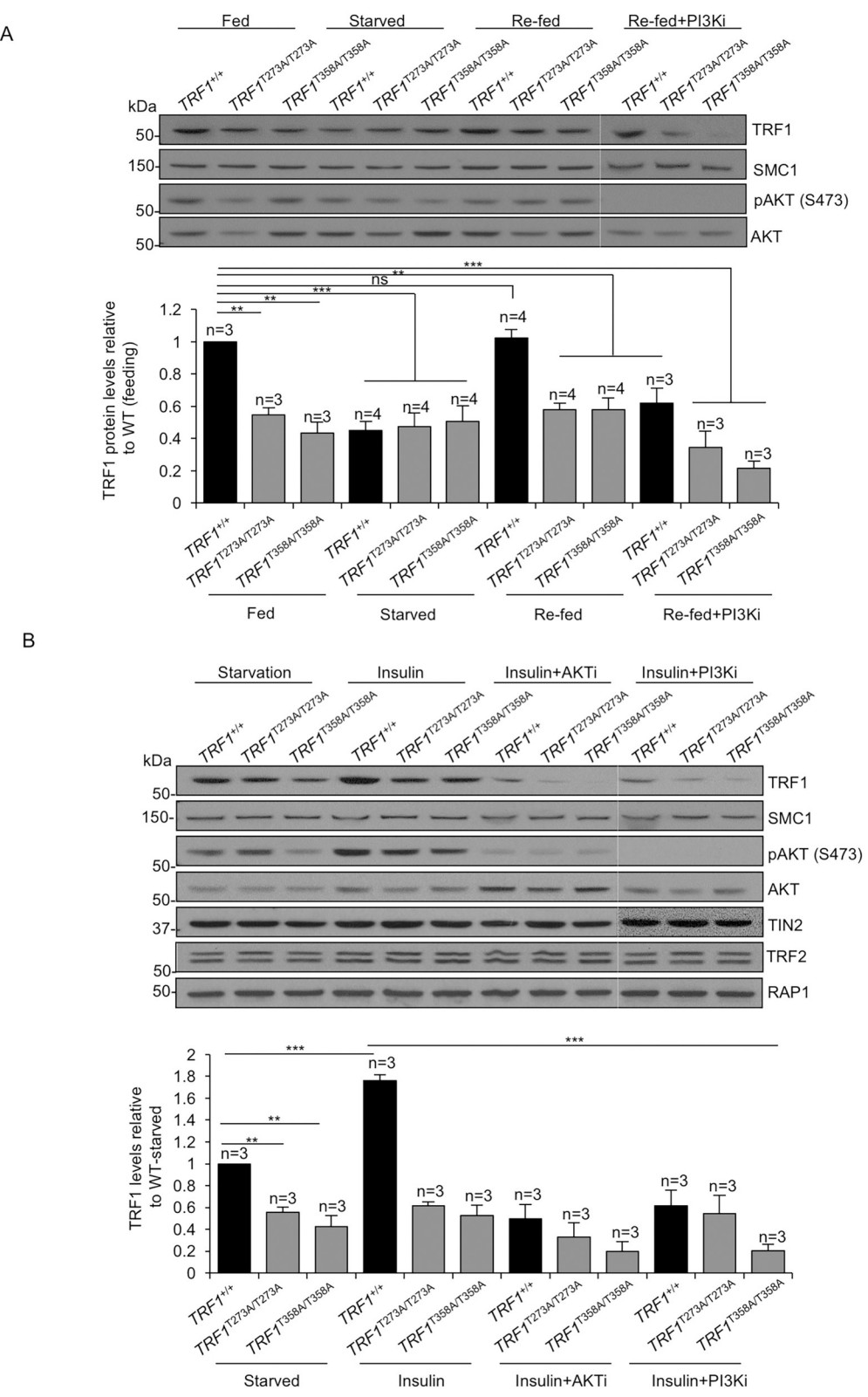

**Fig 7. TRF1 levels respond to extracellular signals in a PI3K/AKT dependent manner. A.** Representative western blot images of TRF1, SMC1, pAKT(S473) and total AKT (top) and quantification (bottom) of TRF1 protein levels in *TRF1*^+/+^, *TRF1*^T358A/T358A^ and *TRF1*^T273A/T273A^ cells under normal fed conditions in DMEM+FBS, starved for 24 hours in DMEM without FBS and either refed for 5 hours with FBS or with FBS plus PI3K inhibitor (ETP47037). **B.**

Representative western blot images of TRF1, SMC1, pAKT(S473), total AKT and TIN2 (top) and quantification (bottom) of TRF1 protein levels in *TRF1*$^{+/+}$, *TRF1*$^{T358A/T358A}$ and *TRF1*$^{T273A/T273A}$ cells starved for 24 hours in DMEM without FBS and either refed for 5 hours with insulin, with insulin plus AKT inhibitor (MK22-06) or with insulin plus PI3K inhibitor (ETP47037). Bars and error bars represent mean values ± SE, n = number of independent experiments. Student's t-test was used for the statistical analysis; *p < 0.05, **p < 0.01, ***p < 0.001 ns, no significant.

**Data**). *TRF1*$^{T358A/T358A}$ and *TRF1*$^{T273A/T273A}$ mutant tumors also presented lower cellularity and larger necrotic areas as compared to wild-type tumors (**Fig 8C**). Interestingly, we did not observed significant differences in apoptosis between wild-type and mutant tumors as indicated by the percentage of AC3 positive cells (**Fig 8C and 8F and S1 Source Data**). As expected, TRF1 immunofluorescence analysis showed a sharp reduction in TRF1 levels of 65% and 82% in *TRF1*$^{T273A/T273A}$ and *TRF1*$^{T358A/T358A}$ mutant tumors, respectively (**Fig 8C and 8G and S1 Source Data**). Finally, telomere Q-FISH analysis showed that *TRF1*$^{T273A/T273A}$ and *TRF1*$^{T358A/T358A}$ tumor cells showed a 25–33% decrease in mean nuclear telomere fluorescence compared to wild-type tumors (**Fig 8C and 8H and S1 Source Data**). Together, these results show that non-phophorylatable by AKT TRF1 mutant cells show a decreased tumorigenic potential.

## Discussion

Telomeres, the protective structures at chromosome ends, are essential for chromosome stability and cell viability [2]. Telomeres can also control gene-expression programs important for metabolism and pluripotency [31–34]. These important functions of telomeres are exerted by the binding of the protective shelterin complex to telomeric DNA repeats. In addition, shelterin can also regulate telomerase access to chromosome ends, thereby also regulating telomere length [4,35]. In spite of these key roles of shelterin, very little is known of the regulation of shelterin function by extracellular signals and environmental or developmental cues.

The shelterin complex encompasses TRF1, TRF2, TIN2, TPP1 and RAP1 [34,36]. TRF1 is essential for shelterin function, as TRF1 genetic deletion has been shown to result in telomere deprotection even in the presence of the rest of the shelterin components [7,8]. TRF1 can be phosphorylated by a number of kinases involved in signaling of extracellular signals, including Polo-like kinase 1 (PLK1), Cyclin B-dependent kinase 1 (CDK1), Casein kinase 2 (CK2), Nek7, and AKT [9,37–42]. In particular, we and others previously showed AKT-dependent TRF1 phosphorylation both in mouse and human cell [9,41], but the *in vivo* relevance these TRF1 post-transcriptional modifications on shelterin function were largely unknown.

Here, we show that AKT dependent TRF1 phosphorylation has important roles *in vivo*. We generated non-phosphorylatable human cell lines for the AKT-dependent TRF1 phosphorylation sites by using the CRISPR-Cas9 technology. We demonstrate that AKT-dependent TRF1 phosphorylation is essential for TRF1 protein stability and TRF1 binding to telomeres, and thus, for preventing a persistent telomere DNA damage signaling and occurrence of telomere aberrations. These telomere defects induced by the TRF1 mutants occurred in the absence of changes in the protein levels of the rest of the shelterin components. Binding of the TRF2 shelterin to telomeres was also normal, thus supporting the notion that abolishing AKT-dependent TRF1 phosphorylation is sufficient to disrupt shelterin function even in the presence of the rest of the shelterin components. Importantly, phosphomimetic TRF1 variants show a significant higher affinity binding to telomeric DNA "in vitro", thus supporting a role for these residues in TRF1 binding to telomeres.

Unexpectedly, we also found that the non-phosphorylatable TRF1 mutant cell lines showed shorter telomeres than control cell lines. The short telomere phenotype could be rescued with

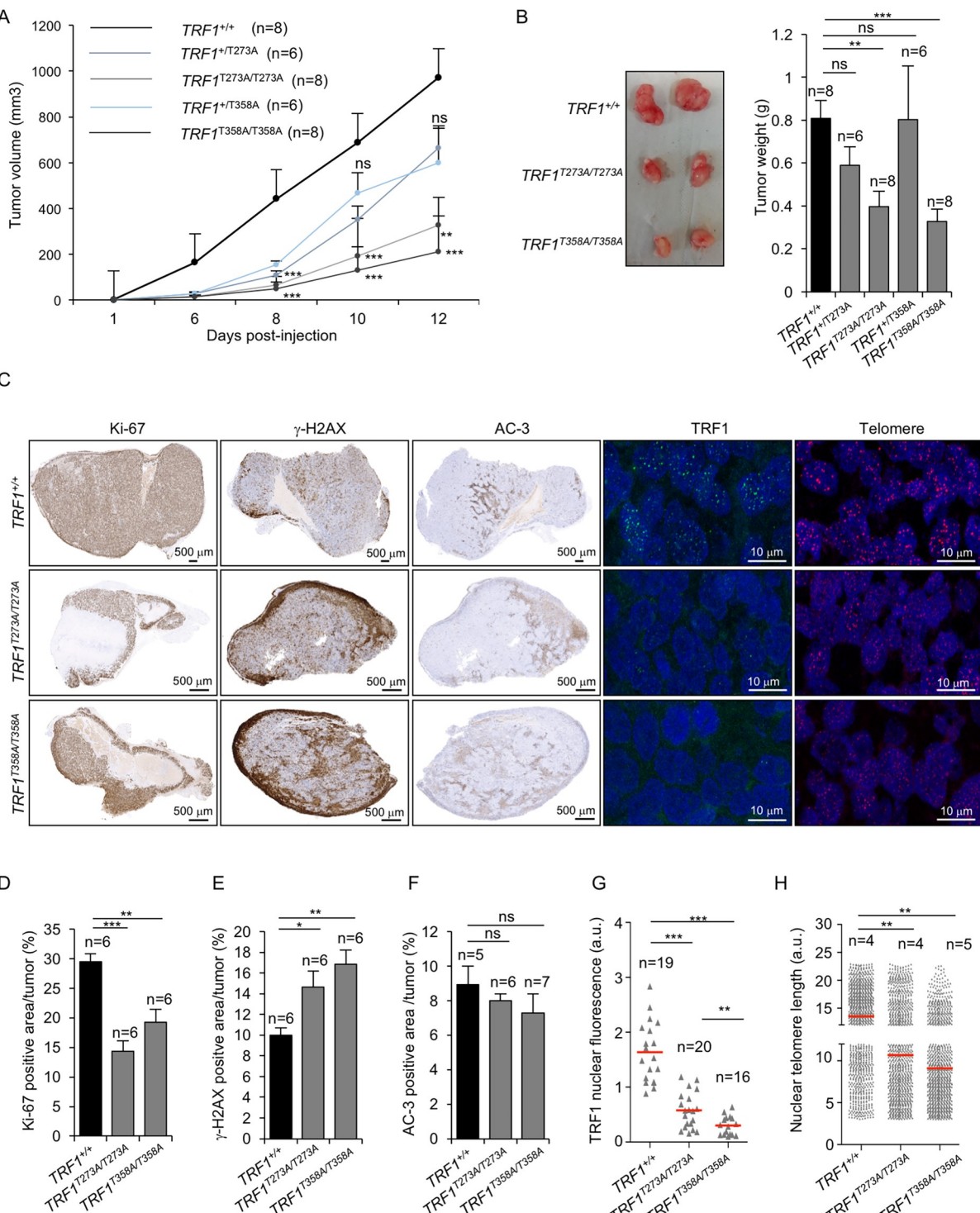

**Fig 8. AKT-dependent TRF1 phosphorylation is essential for tumor growth. A.** Tumor volume follow-up in mice injected with wild type and mutant *TRF1^T358A/T358A^* and *TRF1^T273A/T273A^* HEK293T cells. **B.** Representative image (left) and weight (right) of wild-type and *TRF1^T358A/T358A^* and *TRF1^T273A/T273A^* mutant xenografts. **C-H.** Representative images (C) and quantification of Ki-67 (D), γH2AX (E), Caspase-3 (F), TRF1 (G) and nuclear telomere length (H) of wild-type and *TRF1^T358A/T358A^* and *TRF1^T273A/T273A^* mutant xenografts. Scale bars are indicated. Bars and error bars represent mean values ± SE. n represents number of tumors. Student's t-test was used for the statistical analysis; *p < 0.05, **p < 0.01, ***p < 0.001 ns, no significant.

over-expression of wild-type TRF1 in the homozygous $TRF1^{T273A/T273A}$ mutant cells but not in the $TRF1^{T358A/T358A}$ cells, suggesting that there are different mechanisms responsible for telomere shortening associated to the AKT non-phosphorylatable TRF1 mutants. In the case of $TRF1^{T358A/T358A}$ cells, the fact that TRF1 over-expression cannot rescue the short telomere phenotype suggest that this mutation could have a dominant negative effect. These findings are further supported by the fact that, in contrast to mutant TRF1 proteins in the AKT phosphorylation sites which show short telomeres, a complete abrogation of TRF1 proteins levels by shRNA results in telomere elongation as previously reported [6], suggesting that telomere length defects shown by $TRF1^{T273A/T273A}$ and $TRF1^{T358A/T358A}$ mutant cells are not due to the lower protein stability and consequent reduction in TRF1 protein levels in mutant cells but rather to a *de novo* gain of function of these mutations exerting a dominant negative effect. The fact that TRF1 phosphorylation-defective mutants do not impair telomere lengthening by telomerase and given the functional TRF1 role in telomeric DNA replication [7,8], suggest that the effects of T273A and T358A mutations in telomere shortening are due to a defect of these variants in assisting telomere replication either by impeding replication of telomeres or by enhancing the degree to which replication forks stall and telomere breakage occurs. In support of this, we observed elevated MTS and higher levels of phospho CHK1 in $TRF1^{T273A/T273A}$ and $TRF1^{T358A/T358A}$ mutant cells, indicative of telomeric DNA replicative damage. Future research is needed to study the effect on telomere maintenance of AKT-dependent regulation of TRF1 in ALT cells.

These findings also suggest that TRF1 may be regulated in response to extracellular signals via AKT-dependent TRF1 phosphorylation. Indeed, we show here that TRF1 levels are regulated by extra-cellular signals mediated by the PI3K/AKT pathway such as starvation or exposure to insulin and that TRF1 mutant cells in the AKT-dependent phosphorylation sites show an impaired response to these proliferative extracellular signals. The response to extracellular cues at telomeres seems to be specific for TRF1 since other shelterin component remain unaffected to milieu changes.

Furthermore, in line with this, we also show a decreased tumorigenesis potential of AKT non-phosphorylatable TRF1 mutants, which is coincidental with shorter telomeres and increased DNA damage in the TRF1 mutant tumors. As PI3K mutations and AKT hyper-activation frequently occurs in cancers, our results suggest that these alterations may also favor TRF1 stability, telomere protection and telomere maintenance, a hallmark of cancer cells [20].

## Materials and methods

### Generation of *TRF1* knock-in cell lines by CRISPR-Cas9 system

The sequence for each sgRNA were obtained using an online CRISPR Design Tool [15]. The sgRNA were cloned into the plasmid pSpCas9(BB)-2A-GFP (Addgene,#48138) using a protocol described in [15]. A homologous 100-nt single-stranded DNA oligonucleotides (ssODN) harboring the desired nucleotide substitutions was used as template for homologous recombination repair of the CRISPR-Cas9 cleavage site. The ssODN was designed to change the Cas9 cleavage site to avoid re-cutting and to introduce a new restriction site for genotyping without altering the encoded amino acid sequence [43]. Sequences of sgRNA and ssODN can be found in S1 Table.

### Cells and culture conditions

Human embryonic kidney (HEK) 293T cells were purchased from ATCC and grown in Dulbecco´s modified Eagle´s medium (DMEM) with 10% of fetal bovine serum (FBS) and 1% (vol/vol) penicillin–streptomycin (Gibco). The cells were grown at 37˚C with 5% $CO_2$. Cell

lines were regularly tested for mycoplasma using the Myco Alert Mycoplasma Detection Kit (Lonza). Cells were seeded on 6-well plates at 100,000 cells per ml. At 60% confluence, HEK293T cells were transfected with 2μg Cas9 plasmid and 1μg ssODN using Fugene HD transfection (Promega) following the manufacturers' protocol.

For *TRF1* know-down, cells were transfected with vector expressing short hairpin RNA for TRF1 (*shTRF1*, SHCLNG-NM_017489, SIGMA ALDRICH).

HEK293T cells were grown in high-glucose DMEM supplemented with 10% FBS. In starvation experiments, cells were washed twice with PBS and serum-free high-glucose DMEM was added for 24h. After starvation, cells were either serum (10% FBS) or insulin (10 ng/ml) stimulated for five hours. When indicated MK22-06 (10 μM), an inhibitor of AKT phosphorylation at S473 and T308 (https://www.selleck.eu/products/MK-2206), or PI3K inhibitor ETP47037 (10 μM) was also added.

To generate *TRF1* knock-in HEK293T cells were transiently co-transfected with plasmid pSpCas9(BB)-2A-GFP and specific ssODN. Two days after cells were trypsinized, washed with PBS and re-suspended in FACS solution (0.1% BSA, 3 mM EDTA in PBS). Single cells of the 10% GFP brightest ones were sorted using the FACS ARIA IIU (Becton Dickinson) and plated onto 96-wells plates for monoclonal cell expansion. Genomic DNA was obtained using a DNeasy kit (69504, QIAGEN) and different TRF1 locus fragments were amplified using Phusion High-Fidelity DNA Polymerase (F53OL, Thermo Fisher) and primers positioned outside of the corresponding mutated sequence (S1 Table). Restriction-fragment length polymorphism (RFLP) analysis using PvuI (R0150, New England Biolabs) for T273A mutation or PstI (R0140, New England Biolabs) for T358A mutation was performed for clone identification. Selected clones were further validated by Sanger sequencing. Homozygous and heterozygous monoclonal cell lines were selected.

For iPS cell generation, plasmids pMX-hOCT3/4, pMX-hSox2, pMX-hklf4, pMX-hc-MYC (Addgene #17217, 13367, 13370, 17220, respectively) were transfected into TRF1 wild-type and TRF1 knock-in mutant HEK293T cells.

## Generation of *TRF1* mutant alleles

TRF1 mutant alleles were generated by site-directed mutagenesis using the QuikChange II XL Site-Directed Mutagenesis Kit (200522, Agilent Technologies). Briefly, PCRs were performed following the manufacturer´s protocol with both the pTrcHisB-TRF1 (Addgene #53209) and pLPC-FLAG-TRF1 (Addgene #16058) vectors as template and following PAGE purified mutagenic primers: TRF1273A-F, TRF1273A-R, TRF1344-F, TRF1344-R, TRF1358A-F, TRF1358A-R, TRF1-T273A-T358A, TRF1273D-F, TRF1273D-R, TRF1358D-F, TRF1358D-R. Sequences can be found in S1 Table.

PCR products were digested with DpnI restriction enzyme to digest the parental DNA for 1 h at 37˚C and then, transformed into XL-10-Gold ultracompetent cells. Individual colonies were grown and DNA extracted with QIAprep Spin Miniprep Kit (27106, QIAGEN). Mutations were confirmed by Sanger sequencing with a *TRF1* specific primer TRF1-seq.

## Expression and purification of recombinant His-TRF1

pTrcHisB-TRF1, pTrcHisB-TRF1-T273A, pTrcHisB-TRF1-T344A, pTrcHisB-TRF1-T358A, pTrcHisB-TRF1-T273A-T358A, pTrcHisB-TRF1-T273D, pTrcHisB-TRF1-T358D were expressed into Rosseta (DE3)pLysS *E. coli*. One-liter cultures of *E. coli* transformed with these vectors were incubated for 4 h in the presence of 1 mM isopropyl b-D-1-thiogalactopyranoside (IPTG, Sigma) at 37˚C to induce expression. The cell pellet was resuspended in Lysis Buffer (50 mM $NaH_2PO4$, 300 mM NaCl, 10 mM Imidazole) and then lysed by sonication. Following

centrifugation, the supernatant of 6xHis-TRF1 variants were purified by immobilized metal ion affinity chromatography (IMAC) using a Ni Sepharose 6 Fast Flow (GE Healthcare) for 2 h at 4°C. Then the beads were washed three times with Wash Buffer (50 mM $NaH_2PO4$, 300 mM NaCl, 50 mM Imidazole). The bound proteins were eluted with Elution Buffer (50 mM $NaH_2PO4$, 300 mM NaCl, 300 mM Imidazole) and finally proteins were recovered in 20 mM HEPES pH 7.5, 300 mM NaCl, 10% glycerol and 0.5 mM DTT. The proteins were aliquoted, flash-frozen with liquid nitrogen and stored at -80°C until further use.

### *In vitro* TRF1 phosphorylation

One μg of His-TRF1 was incubated with 0.2 μg of human GST-AKT1 (1379-0000-2, ProQuinase) in kinase buffer (250 mM HEPES (pH 7.5), 50 mM $MgCl_2$, 25 mM DTT and 0.5 mM ATP) containing 5 μCi [γ-$^{32}$P] ATP in a total volume of 25 μl. The reactions were performed at 30°C for 1 h and stopped by addition of Laemmeli buffer. Samples were resolved in 4–12% SDS-PAGE gels and subjected to autoradiography.

### Electrophoretic mobility shift assay (EMSA)

*In vitro* gel-shift assays were performed with 1 μg purified His-TRF1 WT or phosphomimetic mutant variants (T273D and T358D) and $^{32}$P-labeled ds (TTAGGG)7 telomeric probe. Briefly, DNA probes were prepared by annealing the two oligonucleotides, sense (5 ′-GGGTTAGGGTTAGGGTTAGGGTTAGGGT- TAGGGTTAGGGTTAGGGCCCCTC-3′) and antisense (5′-GAGGGGCCC- TAACCCTAACCCTAACCCTAACCCTAACCC- TAACCCTAACCC-3′), end labeled with [γ-$^{32}$P] ATP and T4 polynucleotide kinase (New England Biolabs) and purified by free nucleotide removal spin column (GE Healthcare). Labeled DNA probes were incubated with 1 μg of purified protein for 20 min at room temperature in a 30 μl reaction containing 40 mM HEPES pH 7.9, 300 mM KCl, 10% (v/v) glycerol, 8% (v/v) Ficoll, 2 mM EDTA, 0.2 mM $MgCl_2$, 1 mM DTT, 600 μg of bovine serum albumin and 30 μg of poly (dl-dC) (Sigma-Aldrich). The DNA-protein complexes were resolved on a 5% non-denaturing polyacrylamide (29:1 acrylamide:bisacrylamide) gel with 0.5X Tris-borate-EDTA, TBE (Sigma-Aldrich) as running buffer. Gels were dried under a vacuum at 80°C and autoradiographed.

### Western blot analysis

Nuclear and cytoplasmic protein extracts were obtained using a Nuclear/Cytosolic Fractionation Kit (K266-100, Biovision) and protein concentration was determined using a Bradford Reagent (B6916, Sigma Aldrich). Twenty micrograms of nuclear extracts and forty micrograms of cytoplasmic were separated in 4–12% SDS-PAGE gels (NuPAGE Invitrogen) and transferred to nitrocellulose membranes (Amersham Protan). Blots were incubated with the indicated antibodies. Antibody binding was detected after incubation with a secondary antibody coupled to horseradish peroxidase using a chemiluminescence with ECL detection kit (GE Healthcare). In the protein stability assay, cells were treated with cycloheximide (100 mg/ml) (Sigma-Aldrich) and collected at indicated time points. Relative protein levels in the immunoblots were quantified using ImageJ image analysis software. The protein levels were normalized to SMC1 and the relative levels at time 0 defined as 1.

The primary antibodies used were anti-TRF1 (1:1000) (BED5, Bio-Rad), anti-TIN2 (1:1000) (ab197894, Abcam), anti-POT1 (1:000) (ab124784, Abcam), anti-TRF2 (1:1000) (NB110-57130, Novus Biologicals), anti-RAP1 (1:1000) (A303-532A, Bethyl) and anti-SMC1 (1:8000) (A300-055A, Bethyl), anti phospho-AKT-(Ser473) (1:1000) (9261, Cell Signaling Technology), anti AKT (1:1000) (07–416, Millipore), anti phospho-S6 ribosomal protein

(Ser235/236) (1:1000) (2211, Cell Signaling Technology), anti S6 ribosomal protein (1:1000, Cell Signaling Technology) and anti-FLAG (1:1000) (F3165, Sigma-Aldrich).

## Immunofluorescence and immunohistochemistry analysis

HEK293T cells were plated in Poly-L-lysine-coated coverslips, treated for 5 min with Triton-100 buffer [44] for nuclear extraction, fixed 10 min in 4% buffered formaldehyde, permeabilized with 0.2% PBS-Triton for 10 min and blocked with 5% fetal bovine serum in PBS for 1h. Samples were incubated O/N at 4˚C with anti-TRF1 (1:100) (ab10579, Abcam), anti-RAP1 (1:300) (A300-306A, Bethyl) and anti phospho-histone γH2A.X-Ser139 (1:300) (05–636, Millipore). Cells were then washed and incubated with 488-Alexa or 555-Alexa labeled secondary antibodies (Thermo Fisher Scientific) for 1 h at RT in a humid chamber. Samples were mounted in Prolong Gold with DAPI (Invitrogen). Fluorescent signals were visualized in a confocal ultra-spectral microscope SP5-WLL (Leica). TRF1 and γH2AX signal was quantified using the Definiens Developer XD.2 Software. Image analysis was performed blindly.

Xenografts were fixed in 10% buffered formalin, embedded in paraffin wax and sectioned at 5 mm. Immunohistochemistry staining were performed with primary antibodies against phospho–Histone H2AX (Ser139) (05–636, Millipore), activated-caspase-3 (9661, Cell Signalling) and Ki67 (IR626, Dako). The quantitative analysis of the images was performed blindly using the ZEISS ZEN Microscope Software v2.3.

## Telomeres Q-FISH and chromosomal aberrations on metaphase

HEK293T wild type and mutant cells were incubated with 0.1 μg/ml during 4 h. After hypotonic swelling in 0.03 M sodium citrate for 30 min at 37 ˚C, cells were fixed in methanol:acetic acid (3:1). Quantitative telomere fluorescence in situ hybridization (Q-FISH) was performed as described [45]. Images were captured using microscope Leica DM6B using a 100x oil objective. Telomere length was analyzed using Leica Application Suite X Software. The images were analyzed blindly.

## RNA and qPCR

Total RNA from cells was extracted with the RNeasy kit (74106, QIAGEN) and reverse transcribed was using the iSCRIPT cDNA synthesis kit (1708891, BIO-RAD) according to manufacturer's protocol. Quantitative real-time PCR was performed with the QuantStudio 6 Flex (Applied Biosystens, Life Technologies) using Go-Taq Green Master Mix (M7123, Promega) according to the manufacturer's protocol. All values were obtained in triplicates. Primers used were RT-TRF1-F and RT-TRF1-R. *Alpha Tubulin* was used as the housekeeping gene using primers RT-Tubulin-F and RT-Tubulin-R. Sequences can be found in S1 Table. We determined the relative expression of *TRF1* in each sample by calculating the 2ΔCT value. For each sample, 2ΔCT was normalized to control 2ΔCT mean.

## Telomere Restriction Fragment analysis (TRF)

Cells were isolated and embedded in agarose plugs and digested with MboI (R0147, New England Biolabs). DNA was separated by gel electrophoresis in 0.5X TBE maintained at 14˚C, using a CHEF DR-II pulsed-field apparatus (Bio-Rad) for 12 h at 5 V/cm at a constant pulse time of 5 s. The gel was transfer to nylon membrane (Hybond-XL, GE Healthcare) and probed with a $^{32}$P-labeled telomeric probe (TTAGGG)$_n$ (a gift from T. de Lange). Mean TRF length were determined using ImageQuant TL.

## Co-immunoprecipitation

For immunoprecipitation, cells were washed with PBS and were lysed in 1X cell lysis buffer (9803, Cell Signaling Technology) supplemented with Complete protease inhibitors (Roche). After incubation on ice for 20 min with occasional mixing, the cell lysates were centrifuged at 14,000 rpm for 10 min at 4˚C and supernatants were collected for immunoprecipitation. We prepared lysates from one 10-cm dish and mixed them with 50 μl of a 50% slurry of Flag M2 agarose beads (A2220, Sigma-Aldrich) overnight, immunoprecipitates were washed four times whit lysis buffer and eluted with Laemmli loading buffer.

## ChIP assay and telomere dot-blots

For ChIP analysis, $3 \times 10^6$ of HEK293T cells stably expressed Flag-TRF1, Flag-TRF1-T273A and Flag-TRF1-T358A cells were used per condition. Formaldehyde was added directly to culture medium to a final concentration of 1% and incubated for 15 min at room temperature (RT) on a shaking platform. Cross-linking was then stopped by addition of glycine to a final concentration of 0.125 M for 5 min at RT. Cross-linking cells were washed twice with cold PBS containing 1 μM PMSF and protease inhibitors and then pelleted. Cells were lysed in lysis buffer (1% SDS, 10 mM EDTA and 50 mM Tris-HCl pH 8.0) containing protease inhibitors at $2 \times 107$ cells/ml for 20 min at 4˚C. Lysates were sonicated to obtain chromatin fragments <1 kb and centrifuged for 15 min in a microfuge at room temperature. Chromatin was diluted 1:10 with dilution buffer (1.1% Triton X-100, 2 mM EDTA pH 8.0, 150 mM NaCl, 20 mM Tris-HCl pH 8.0) contained protease inhibitors and precleared with 50 μl of protein A/G Plus-Agarose beads (sc-2003, Santa Cruz Biotechnology). After centrifugation, chromatin fragments were incubated with one of the following at 4˚C overnight on a rotating platform: anti-FLAG M2 (F1804, Sigma-Aldrich), anti TRF2 (NB110-57130, Novus Biologics) or normal rabbit IgG (sc-2025, Santa Cruz Biotechnology). Samples were then immunoprecipitated with 50 μl of protein A/G Plus-Agarose beads were added and incubated for 2 h. The immunoprecipitated pellets were washed once with IP Wash A (0.1% SDS, 1% Triton X-100, 2 mM EDTA, 20 mM Tris-HCl pH 8.0), and IP Wash B (150 mM NaCl, and then with 0.1% SDS, 1% Triton X-100, 2 mM EDTA, 20 mM Tris-HCl pH 8.0), and IP Wash C (500 mM NaCl, and next with 0.25 M LiCl, 1% Nonidet P-40, 1% sodium deoxycholate, 1 mM EDTA, and 10 mM Tris-HCl pH 8.0) and finally with TE (10 mM Tris-HCl pH 8.0 and 1 mM EDTA) two times. The chromatin was eluted from the beads twice by incubation with 250 μl 1% SDS and 0.05 M NaHCO$_3$ during 15 min at RT with rotation. After adding 20 μl of 5 M NaCl, the crosslink was reversed overnight at 65 ˚C. Samples were supplemented with 20 μl of 1 M Tris-HCl pH 6.5, 10 μl of 0.5 M EDTA, 20 μg of RNase A, and 40 μg of proteinase K, and were incubated for 1 h at 45 ˚C. DNA was recovered by phenol–chloroform extraction and ethanol precipitation, denatured (0.3 M NaOH) at 50 ˚C for 1 h, neutralized (1 M Ammonium acetate), and transferred to a Hybond-N+ membrane (Amersham) on a dot blot, and hybridized with a telomeric probe obtained from a plasmid containing 1.6 kb of TTAGGG repeats (gift from T. de Lange, Rockefeller University) or with ALU sequence probe. The signal was quantified with the ImageJ software. The amount of telomeric DNA immunoprecipitated was calculated in each ChIP based on the signal relative to the corresponding total telomeric DNA signal.

## Xenografts experiments

For xenografts experiments, 6-week athymic nude females were obtained from Harlan (Foxn1$^{nu/nu}$). Mice were maintained at the Spanish National Cancer Centre (CNIO) in accordance with the recommendations of the Federation of European Laboratory Animal Science Associations (FELASA) under specific pathogen-free conditions. All animal experiments were

performed in accordance with the guidelines stated in the International Guiding Principles of Biomedical Research Involving Animals, developed by the Council for International Organizations of Medical Sciences (CIOMS), and were approved by Ethical Committee (CEIyBA) (IACUC.003-2015, CBA-032015). Along with those guidelines, mice were monitored in a daily or weekly basis and they sacrificed in $CO_2$ chambers when the human end-point was considered.

Wild type and mutants HEK293T cells were dissociated and re-suspended in DMEM and Matrigel in a 1:1 ratio. Foxn1$^{nu/nu}$ mice were subcutaneously injected with 4 x $10^6$ cells in 100 ul of DMEM:Matrigel. Mice were weighted and tumors were measured every 2–4 days. Tumor volume was determined by the following equation: V = a*b$^2$, were a and b are tumor length and width respectively.

## Supporting information

**S1 Table. Primers used in this study.**
(DOC)

**S1 Fig. Purity and specificity of 6His-tagged TRF1 variant purification from *E. coli*. A-B.** Coomassie stained SDS-PAGE gel (A) and western blot (B) of affinity purified His-TRF1-WT, His-TRF1-T273A, His-TRF1-T344A and His-TRF1-T358A (2μg). The MW ladder is shown to the left.
(TIFF)

**S2 Fig. Characterization of independent mutant clones. A-B.** Representative western blot images of total nuclear TRF1 protein levels in wild type and in different independent heterozygous and homozygous clones. The heterozygous mutant *TRF1$^{+/T273A}$* and *TRF1$^{+/358A}$* clones correspond to C2 & C8 and to C7 & C13 & C16, respectively. The homozygous *TRF1$^{T273A/T273A}$* and *TRF1$^{T358AT/358A}$* mutant clones correspond to C9 & C11 and to C5 & C1, respectively. Those clones used throughout the manuscript are labeled in red. The fold change in TRF1 levels with regards to wild type cells is indicated below the images. SMC1 was used as a loading control. **C.** Quantification of *TRF1* transcriptional levels by q-PCR and TRF1 protein levels in *TRF1$^{+/+}$* cells transfected with an sh-*TRF1*. SMC1 was used as a loading control. A representative western blot images of total nuclear TRF1 protein levels is shown in right panel. Error bars represent standard deviation. *n* number of independent experiments. Student's t test was used for statistical analysis, P values are shown. ***, p $\leq$ 0.001.
(TIFF)

**S3 Fig. Non-phosphorylatable *TRF1 knock-in* cell show a DDR activation. A-B** Representative western blot images and quantification of phosphor-CHK1 (A) and phosphor-CHK2 (B) with regards to total CHK1 and total CHK2 in wild-type and homozygous mutant cells. SMC1 was used as loading control. Error bars represent standard deviation. *n* number of independent experiments. Student's t test was used for statistical analysis, P values are shown. *, p $\leq$ 0.05; **, p $\leq$ 0.01; ***, p $\leq$ 0.001.
(TIFF)

**S4 Fig. Proliferation ability. A.** Growth rate of *TRF1$^{+/+}$*, *TRF1$^{+/T273A}$*, *TRF1$^{T273A/T273A}$*, *TRF1$^{+/T358A}$* and *TRF1$^{T358A/T358A}$* cell lines. Student's t test was used for statistical analysis, p $\leq$ 0.05; **, p $\leq$ 0.01; ***, p $\leq$ 0.001. Error bars represent ± SE. n number of independent experiments.
(TIFF)

**S5 Fig. Telomere length in independent mutant clones.** Telomeric restriction fragment (TRF) blot of *TRF1^T273A^* and *TRF1^T273A^* independent knock-in clones. HE and HO refer to heterozygous and homozygous clones, respectively. Numbers refer to molecular weight standards in Kb. Those clones used throughout the manuscript are labeled in red.
(TIFF)

**S6 Fig. Telomere length effects of TRF1 depletion and Telomerase over expression. A.** Representative western blot images of total nuclear TRF1 protein levels in *TRF1^+/+^*, *TRF1^T273A/T273A^* and *TRF1^T358A/T358A^* transfected with FLAG-TRF1. **B.** Quantification of *TRF1* transcriptional levels by q-PCR in *TRF1^+/+^*, *TRF1^T273A/T273A^* and *TRF1^T358A/T358A^* cells transfected with an sh-*TRF1*. **C.** Representative western blot images of total nuclear TRF1 protein levels in *TRF1^+/+^*, *TRF1^T273A/T273A^* and *TRF1^T358A/T358A^* transfected with an sh-*TRF1*. **D.** Representative Q-FISH images of metaphases spreads and mean telomere length quantification from *TRF1^+/+^*, *TRF1^T273A/T273A^* and *TRF1^T358A/T358A^* cell lines transfected with an sh-*TRF1* at passage 14. Scale bars, 5μm. Student's t test was used for statistical analysis, p ≤ 0.05; **, p ≤ 0.01; ***, p ≤ 0.001. Error bars represent ± SE. n number of metaphases. **E.** Quantification of *TERT* transcriptional levels by q-PCR in *TRF1^+/+^*, *TRF1^T273A/T273A^* and *TRF1^T358A/T358A^* cells transfected pBABE-TERT. Student's t test was used for statistical analysis, p ≤ 0.05; **, p ≤ 0.01; ***, p ≤ 0.001. Error bars represent ± SE. n number of independent experiments.
(TIFF)

**S7 Fig. Telomere length dynamics during reprogramming A.** Quantification of *NANOG* transcriptional levels by q-PCR in *TRF1^+/+^*, *TRF1^T273A/T273A^* and *TRF1^T358A/T358A^* cells transfected with the four Yamanaka's factors. B. Quantification of *TRF1* transcriptional levels by q-PCR in *TRF1^+/+^*, *TRF1^T273A/T273A^* and *TRF1^T358A/T358A^* cells transfected with the four Yamanaka's factors (iPS). Student's t test was used for statistical analysis, p ≤ 0.05; **, p ≤ 0.01; ***, p ≤ 0.001. Error bars represent ± SE. n number of independent experiments. **C.** Representative Q-FISH images of metaphases spreads and quantification of the mean telomere length in *TRF1^+/+^*, *TRF1^T273A/T273A^* and *TRF1^T358A/T358A^* cells transfected either with the four Yamanaka's factors (iPS). n, number of metaphases. Student's t test was used for statistical analysis, *, p ≤ 0.05; **, p ≤ 0.01; ***, p ≤ 0.001. Error bars represent the SE. D. Representative image of telomeric restriction fragment (TRF) blot of *TRF1^+/+^*, *TRF1^T273A/T273A^* and *TRF1^T358A/T358A^* cell lines transfected with the four Yamanaka's factors (iPS) at progressive passages. Numbers refer to molecular weight standards in Kb. Mean telomere length for each cell line is shown at the base of the lanes.
(TIFF)

**S1 Source Data. Numerical data that support the findings of this study.**
(XLSX)

## Acknowledgments

We thank D. Megias from his support in microscopy analysis and R. Serrano for her support in xenograft experiments.

## Author Contributions

**Conceptualization:** Maria A. Blasco.

**Data curation:** Raúl Sánchez-Vázquez, Paula Martínez, Maria A. Blasco.

**Formal analysis:** Raúl Sánchez-Vázquez, Paula Martínez, Maria A. Blasco.

**Funding acquisition:** Maria A. Blasco.

**Investigation:** Raúl Sánchez-Vázquez, Maria A. Blasco.

**Methodology:** Raúl Sánchez-Vázquez, Paula Martínez, Maria A. Blasco.

**Project administration:** Maria A. Blasco.

**Resources:** Maria A. Blasco.

**Supervision:** Maria A. Blasco.

**Validation:** Maria A. Blasco.

**Visualization:** Maria A. Blasco.

**Writing – original draft:** Raúl Sánchez-Vázquez, Paula Martínez, Maria A. Blasco.

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
