## [Decision Letter · Decision Letter 0]

25 Jan 2021

Dear Dr Blasco,

Thank you very much for submitting your Research Article entitled 'AKT-dependent signaling of extracellular cues through telomeres impact on tumorigenesis' to PLOS Genetics.

The manuscript was fully evaluated at the editorial level and by independent peer reviewers. The reviewers appreciated the attention to an important topic but identified some concerns that we ask you address in a revised manuscript.

We therefore ask you to modify the manuscript according to the review recommendations. Your revisions should address the specific points made by each reviewer.

[LINK]

Yours sincerely,

Matthew L. Meyerson, M.D., Ph.D.

Guest Editor

PLOS Genetics

Gregory Barsh

Editor-in-Chief

PLOS Genetics

Reviewer's Responses to Questions

**Comments to the Authors:**

Reviewer #1: Sanchez-Vazquez and colleagues provide evidence the Akt mediated phosphorylation of telomeric shelterin proteins TRF1 regulates cell growth an in vivo tumorigenesis. This report expands on previous work from this group where the Akt phosphorylation sites on mouse TRF1 were identified (Nat Commun 2017). The data are well controlled, using a range of in vitro and in vivo methods. I review a lot of papers and I must say that this manuscript was a pleasure to read-clearly written and not interpreted. I have the following minor comments:

1. Fig. 1 the consensus sequence for Akt phosphorylation is RxRxxS/T-hydrophobic amino acid. To my eyes, the best of the 3 potential sites is the Thr273-perhaps the authors could add a few lines to refer to this in their report?

2. Fig. 1E-with such small error bars, why were the experiments completed 8 times (n=8)? Usually n=3-4 is sufficient? can the authors confirm that n=8 means 8 independent experiments, and if so, how many replicates were included in each independent experiment. The new Superplot method of presenting data should be considered to allow the reader to see all date points and better interpret the data https://rupress.org/jcb/article/219/6/e202001064/151717/SuperPlots-Communicating-reproducibility-and)

3. Fid. 2D there does not appear to be a difference between the TRF1+/T273 v TRF1 T273A/T273A mutants in terms of protein levels (in contrast to TRF1T358A mutant)-can the authors comment on this please?

4. Fig. 2D, G-the authors normalise the expression level to the DMSO control-however, it is standard to normalise to the loading control (SMC1 in this case)-please revise.

5. Fig. 2F, 4, 5B, 8C-how were these images quantified and please confirm that images were blinded prior to quantification to avoid conscious and unconscious bias.

6. Fig. 8 A-replot the tumor weight using a similar format used in Fig. 8G to show the proper scatter of the data.

7. For discussion-many groups use pSer473 as a proxy for Akt phosphorylation; however, phosphorylation on Thr308 is also required for full activation. This should be included in future studies. Also, the use of the Akt chemical inhibitor MK22-06 is interesting as it clearly affects TRF1 phosphorylation but I am confused as to why it would also block pSer473 phosphorylation, as this site is phosphorylated by mTORC1 and DNA-PK.

8. For discussion-the TRF1 T344 site did not appear to be an Akt phospho site, with the T344A mutation having no affect on TRF1 levels (Fig. 1E). To me, this would have been a perfect control for any off-target/voodoo effects of CRISPR knock-in cell lines. Did the authors make the TRF1-T344A knockin cell line and show no effects on TRF1 stability/cell proliferation/tumor size etc?

Reviewer #2: This work from the Blasco lab investigated the role of PI3K/AKT in regulating TRF1, a telomere binding protein. They identified 2 distinct targets of AKT-dependent phosphorylation (T273 & T358) in TRF1, which destabilized and perturbed the TRF1 accumulation at telomeres. The mechanism of this was linked with elevated degradation – as the reduced levels were restored by bortezimib, an inhibitor of the proteosome. The stable expression of phospho-TRF1 mutants, by knock-in of 293T cells, elicited telomere damage, accelerated telomere shortening and induced fragility suggestive of replicative complications. While the telomere shortening affect of TRF1-273 mutant cells was rescued by over-expression of wild-type TRF1, the T358 mutant did not, indicative of its dominant negative effect. However, depletion of TRF1 indicated the contrary, that the telomere shortening effect was attributed to a gain of function for TRF1. To pursue what this could be the authors tested a role for these mutants in inhibiting telomerase activity, which was not the case, as tested by ectopic expression of telomerase and through induced pluripotent stem cell generation. The authors tested whether these mutants altered the stability and binding of additional subunits of Shelterin complex. Again, an adverse affect on Shelterin, aside from TRF1 itself, was not evident.

Pursuing the metabolic link of PI3K/AKT, the authors finally examined a role for TRF1 and these phosphorylations in response to extracellular sensing of insulin and starvation. Whereas cells expressing wild-type TRF1 significantly induce the expression of TRF1 protein, cells expressing TRF1-273 or 358 did not respond. Lastly, cells expressing non-phosphorylatable TRF1 alleles did not exhibit nearly as much expansion upon xenograft transplantation. Thus, the authors conclusion is that the phosphorylation of these sites by AKT is required for the transduction of extracellular signaling to telomeres and maintain the tumorigenic proliferation of cancer cells.

The dissection of TRF1 stability and telomeric length/integrity are solid. The only suggestion is to clarify for readers the seeming contradictory statements related to the dominant negative effects. The authors state rather definitively that the phospho-mutants are not dominant negatives before changing this based on the outcomes of the shTRF1 experiments. It would be helpful to perhaps restate or rephrase this section of the text.

A minor point is that the methods do not include a description of how the IPS cells were generated and characterized, or maybe I missed this?

Overall, the paper is pretty comprehensive and its conclusions seem to be on point – albeit somewhat speculative. However, it should be a nice addition to the body of work on TRF1 and potential links with tumorigenesis.

**Have all data underlying the figures and results presented in the manuscript been provided?**

Reviewer #1: Yes

Reviewer #2: Yes

PLOS authors have the option to publish the peer review history of their article (what does this mean?). If published, this will include your full peer review and any attached files.

Reviewer #1: No

Reviewer #2: No

---

## [Editor Report · Decision Letter 1]

9 Feb 2021

Dear Dr Blasco,

We are pleased to inform you that your manuscript entitled "AKT-dependent signaling of extracellular cues through telomeres impact on tumorigenesis" has been editorially accepted for publication in PLOS Genetics. Congratulations!

Yours sincerely,

Matthew L. Meyerson, M.D., Ph.D.

Guest Editor

PLOS Genetics

Gregory Barsh

Editor-in-Chief

PLOS Genetics

Comments from the reviewers (if applicable):

**Data Deposition**

http://datadryad.org/submit?journalID=pgenetics&manu=PGENETICS-D-20-01697R1

**Press Queries**

---

## [Editor Report · Acceptance letter]

23 Feb 2021

PGENETICS-D-20-01697R1 

AKT-dependent signaling of extracellular cues through telomeres impact on tumorigenesis 

Dear Dr Blasco, 

We are pleased to inform you that your manuscript entitled "AKT-dependent signaling of extracellular cues through telomeres impact on tumorigenesis" has been formally accepted for publication in PLOS Genetics! Your manuscript is now with our production department and you will be notified of the publication date in due course.

With kind regards,

Alice Ellingham

PLOS Genetics

On behalf of:
